# Changes in Physicochemical and Microbiological Properties, Fatty Acid and Volatile Compound Profiles of Apuseni Cheese during Ripening

**DOI:** 10.3390/foods10020258

**Published:** 2021-01-27

**Authors:** Crina Carmen Mureşan, Romina Alina (Vlaic) Marc, Cristina Anamaria Semeniuc, Sonia Ancuţa Socaci, Anca Fărcaş, Dulf Fracisc, Carmen Rodica Pop, Ancuţa Rotar, Andreea Dodan, Vlad Mureşan, Andruţa Elena Mureşan

**Affiliations:** 1Department of Food Engineering, Faculty of Food Science and Technology, University of Agricultural Sciences and Veterinary Medicine of Cluj-Napoca, 3-5, Mănăştur Street, 400372 Cluj-Napoca, Romania; crina.muresan@usamvcluj.ro (C.C.M.); cristina.semeniuc@usamvcluj.ro (C.A.S.); andreea-doina.dodan@usamvcluj.ro (A.D.); vlad.muresan@usamvcluj.ro (V.M.); andruta.muresan@usamvcluj.ro (A.E.M.); 2Department of Food Science, Faculty of Food Science and Technology, University of Agricultural Sciences and Veterinary Medicine of Cluj-Napoca, 3-5, Mănăştur Street, 400372 Cluj-Napoca, Romania; sonia.socaci@usamvcluj.ro (S.A.S.); anca.farcas@usamvcluj.ro (A.F.); carmen-rodica.pop@usamvcluj.ro (C.R.P.); anca.rotar@usamvcluj.ro (A.R.); 3Faculty of Agriculture, University of Agricultural Sciences and Veterinary Medicine Cluj-Napoca, Cluj-Napoca 400372, 3-5 Calea Mănăştur Street, 400372 Cluj-Napoca, Romania; francisc.dulf@usamvcluj.ro

**Keywords:** matured cheese, nutrients, lipids, aroma compounds

## Abstract

The evolution during ripening on the quality of Apuseni cheese was studied in this research. The cheese samples were controlled and evaluated periodically (at 4 months) during 16 months of storage (at 2–8 °C) for physicochemical parameters (pH, moisture, fat, fat in dry matter, total protein, ash, NaCl), microbiological (total combined yeasts and molds count (TYMC), total viable count (TVC), *Escherichia coli*, *Staphylococcus*
*aureus*, *Salmonella*, lactic acid bacteria (LAB)), fatty acids (FA) and volatile compounds. For better control of the quality of the cheese, the storage space was evaluated for TYMC and TVC. The ripening period showed improved effects on the quality of the cheese, showing lower values for moisture and pH and an increase in macronutrients. Both the cheese samples and the storage space were kept within the allowed microbiological limits. Lipids are predominant, the predominant FAs being saturated fatty acids (SFAs), which decrease, while monounsaturated fatty acids (MUFAs) increase. During ripening, the microbiological and chemical changes result in the development of flavor. Major volatile compounds such as 2-heptanone show accumulations, while acetophenone, limonene, or thymol show a decrease. In conclusion, Apuseni ripening cheese clearly involves a complex series of transformations, leading to a ripening cheese with improved nutritional and aromatic characteristics.

## 1. Introduction

The name “cheese” is used for products obtained from fresh or ripened coagulated milk [1]. Several cheese categories belong to the denomination “cheese”. They differ mainly from their moisture content [2]. The assortment range of cheeses is very wide all over the world. In the beginning, cheese was produced to increase the shelf life of milk, but now, cheese is purchased for its sensory and nutritional qualities [3]. Cheese is a food rich in nutritional components, thus constituting an important source of proteins, fatty acids, minerals, and vitamins. Therefore, it is one of the most consumed foods worldwide due to biologically active substances [4].

According to Eurostat, in 2018, 156.8 million tons of whole milk was processed in the European Union (EU). EU dairies produced 10.3 million tonnes of cheese in 2018 and 10.2 million tons of cheese in 2017. Over 90% of the cheese was produced from pure cow’s milk, with 2% from pure ewes ’or pure goats’ milk. Fresh cheese represented the largest share of total EU cheese production (34%, or 3.5 million tons of cheese), followed by medium hard cheese (26%, or 2.7 million tons) and hard cheese (19%, or 1.9 million tons). Cheese production in Romania has increased in the last 10 years from 66,290 tons to 96,400 tons, ranking 18th in the EU [2].

This upward trend is set to continue in the coming years, unlike other European Union countries where both the production and consumption of cheese have stabilized. One of the underlying factors of this growth was the development of small local factories, new varieties of cheese, as is our case, as well as increasing market demand for matured cheeses. Although in recent years, there has been a growing interest in studying the characteristics of cheeses, not enough research has been conducted, due to the diversity of the range. However, as far as we know, no studies have been performed on the chemical composition or the main biochemical changes during the maturation of Apuseni cheese.

Apuseni cheese is a type of hard cheese made in the Apuseni Mountains (Colteşti, Romania) from pasteurized cow’s milk, from small productions in the area, ripening up to 16 months. This variety of cheese is a local one, which is marketed in different stages of ripening, being highly appreciated by consumers. It has a parallelepiped shape, weight 7 kg, height 7–10 cm, length 25 cm, and width 25 cm. It is recognized by the holes formed in the process of decomposition of lactic acid and glutamic acid, when existing microorganisms produce carbon dioxide that remains trapped in the dense mass of cheese. The taste is specific, slightly sweet, and due to its propionic ferments. The skin is yellow, the core is yellow, and it is uniform throughout the table, with a slightly more intense luster at the meshes.

Although various studies have been carried out in recent years to characterize ripening cheeses, they have been carried out to a greater or lesser extent, such as Parmigiano Reggiano [5], Cheddar [6], Évora Cheese [7], Terrincho chees [8], Gorgonzola cheese [9], Civil cheese [10], and Spanish blue cheese-Valdeón [11].

Starting from the existing study, some authors report changes in the chemical composition (protein and polyunsaturated fatty acids (PUFA)) in cheese due to heat treatments [12,13], while more recent studies have shown that heat treatment does not affect the amount of heat, protein, or cheese fatty acid profile [14]. Claeys et al. conclude that present or added microorganisms during cheese making can release bioactive compounds [15].

The composition of the milk, the technological process, and the ripening time for curd cheeses can last between 2 weeks (Mozzarella) and 2 or more years (Cheddar, Parmigiano-Reggiano), during which biological and microbiological changes take place, resulting in flavors specific to each cheese assortment. These changes are based on the interactions between starter bacteria, enzymes from milk and rennet and lipases [12,13].

As a general rule, factors that increase the rate of ripening increase the risk of flavor development and reduce the period of time when the cheese is saleable. Protein degradation during cheese curing is a directed process resulting in protein fragments with desirable flavors. Dairy fat is a wonderfully rich source of flavors, because it contains an extremely diverse selection of fatty acids [7,13,16]. Following the cheese-ripening processes, changes were observed in the physical, chemical, and microbiological properties [5,8,13,17].

In our view, one of the ways to protect these qualities in local products, such as Apuseni cheese, is to carry out a global characterization to determine the main characteristics that make them original, specific, and distinctive products on the market. Consequently, the aim of our study was to analyze the physicochemical, microbiological, and biochemical changes during the maturation of Apuseni cheese, paying special attention to fatty acids and flavor compounds.

## 2. Materials and Methods

### 2.1. Materials

The Apuseni cheese samples (Figure 1) were made according to the flow diagram presented in Figure 2 and were taken from a milk processing unit in Romania (Coltești dairy products). The cultures used were acidification crops and Cryofast cultures, coagulation power 1800 IMCU/g, produced by Company of Sacco System, Italy. The microbial enzymatic clot (Rhizomucor miehei) was produced by Ideal Still Exim Srl.

The ripening of the samples took place over a period of 16 months. Periodically, the peel was washed with salt water and, if the peel was too thick, it was scraped. The turning prevented the cheese from deforming, ensuring a uniform maturation throughout the table. The cheeses were turned by hand. Sampling was carried out over several periods, even in different seasons, in order to have continuity and be able to tell the true quality of this type of hard cheese: first sampling: D1–0 months of ripening; second sampling: D2–4 months of ripening; third sampling: D3–8 months of ripening; fourth sampling: D4–12 months of ripening; fifth sampling: D5–16 months of ripening. Samples were collected from 5 different blocks with a scabbard, at each collection, from the same batch, which were homogenized and analyzed in duplicate. The shape of the blocks is parallelepiped, weight 7 kg, height 7–10 cm, length 25 cm, and width 25 cm.

### 2.2. Proximate Composition

Cheese samples were taken when fresh (zero ripening time) on a four-monthly basis during the ripening period and analyses for moisture, ash, salt, total nitrogen, and pH according to Association of Official Analytical Chemists (AOAC, 2000). The pH meter used was Consort C535 pH meter (Consort nv, Turnhout, Belgium). Determination of fat content was performed using the Van Gulik method described by ISO 3433:2008 [18]. Total protein content was determined using the Kjeldahl method described in ISO 8968-1:2014 [19].

### 2.3. Microbiological Analyses

#### 2.3.1. Determination of Total Combined Yeasts and Molds Count (TYMC)

First, 0.1 mL of the diluted sample (10^−2^ and 10^−3^) was transferred to a sterile Petri dish covered with Dichloran-rose bengal chloramphenicol agar (DRBC) (Oxoid, Basingstoke, UK) and spread using a Drigalsky spatula. The plates were incubated for 5 days at 25 °C and counted according to ISO 21527-1: 2008 standard [20].

#### 2.3.2. Determination of Total Viable Count (TVC)

An amount of 5 g cheese sample was weighed into a sterile stomacher bag using sterile instruments. Forty-five milliliters of peptone saline diluent (Oxoid Ltd., Basingstoke, Hampshire, UK) were added and homogenized in a stomacher (Bag Mixer 100 MiniMix, Interscience, St. Nom, France) for 1 min to prepare a 10^−1^ homogenate. Serial dilution for cheese samples was performed in the same conditions up to 10^−8^.

One milliliter of each dilution (10^−3^ and 10^−4^) was aseptically transferred to a sterile Petri dish using a sterile pipette. Into each Petri dish were poured 15 mL of Plate Count Agar (Oxoid Ltd., Basingstoke, Hampshire, UK). Petri dishes were incubated at 30 °C for 48–72 h under aerobic conditions and counted according to ISO 4833:2003 [21].

#### 2.3.3. Quantification of Viable Airborne Microorganisms

The number of airborne microorganisms of cheese-ripening areas was evaluated by culture settling plate method according ISO 4833-2:2013 [22] and ISO 21527-2:2008 [23].

Total viable count (TVC) was collected using Plate Count Agar (Oxoid Ltd., Basingstoke, Hampshire, UK), and the plates were incubated at 37 °C for 72 h for mesophilic bacteria. Total combined yeasts and molds count (TYMC) were collected on Dichloran Glycerol Agar (Oxoid, Basingstoke, UK), and the cultures were incubated at 25 °C for 5 days.

#### 2.3.4. Identification of *Escherichia coli*

The presence of *E. coli* was determined according to ISO 16649-2:2007 [24] standard. The same serial dilution was performed as for other determinations (TVC, TYMC, lactic acid bacteria (LAB), and *Staphylococcus aureus*).

First, 1.0 mL of the diluted sample (10^−1^ and 10^−2^) was uniformly distributed into a sterile Petri dish; then, TBX Agar (Oxoid, Basingstoke, UK) was poured and mixed. The inoculated dishes were inverted and incubated at 44 °C for 24 h. After incubation, the typical colonies of *ß*-glucuronidase positive *E. coli* were counted in each dish (those containing less than 150 typical colonies and less than 300 total colonies) using a colony counter (Colony Star 8500, Funke Gerber, Berlin, Germany). Typical colonies are blue. Non-typical colonies are pale blue. Non-typical colonies found in a Petri dish were also counted and taken into consideration during calculation.

#### 2.3.5. Identification of *Staphylococcus aureus*

ISO 6888-2/A-1/2005 [25] standard method was used. Briefly, 0.1 mL of the diluted sample (10^−2^ and 10^−3^) was transferred to a sterile Petri dish covered with Baird–Parker agar (Oxoid, Basingstoke, UK) supplemented with Egg Yolk Tellurite Supplement (SR 00540, Oxoid, Basingstoke, UK) and spread using a Drigalsky spatula. The plates were incubated for 24–48 h at 37 °C. Typical colonies of coagulase-positive *Staphylococcus aureus* were counted after 24 and 48 h.

#### 2.3.6. Detection of *Salmonella* Species

Twenty-five grams of cheese were suspended in 225 mL of buffered peptone water (Laboratorios Conda, Madrid, Spain) and incubated for 18 h at 37 °C for pre-enrichment. One milliliter of pre-enrichment broth was transferred to 10 mL of RVS (Rappaport-Vassiliadis Soya Peptone) broth (Oxoid Ltd., Basingstoke, Hampshire, UK) and incubated at 42 °C for 24 h. A loopful of the selective enrichment culture was transferred by streaking onto XLD (Agar- Xylose Lysine Deoxycholate) agar (Oxoid Ltd., Basingstoke, Hampshire, UK). The plate was incubated at 37 °C for 24 h. ISO 6579:2002 [26] standard method was used.

After 24 h, the selective agar plate was examined for typical and atypical colonies of *Salmonella*, and biochemical and serological identification tests were performed.

#### 2.3.7. Determination of Lactic Acid Bacteria (LAB)

First, 1.0 mL of the diluted samples (10^−7^ and 10^−8^) was uniformly distributed into a sterile Petri dish; then, Man Rogosa and Sharp (MRS) Agar (Himedia, Mumbai, India) was poured and mixed. The inoculated dishes were inverted and incubated at 37 °C for 48 h under anaerobic condition using the Gas-Pak system (GENER box anae indicator Biomerieux, Nürtingen bioMérieux Deutschland GmbH, Weberstrasse, Nürtingen, Germany) [27].

### 2.4. Determination of Fatty Acid Composition in Cheese

The total lipids of the samples (10 g) were extracted using chloroform: methanol mixture [28]. Fatty acid (FA) methyl esters (FAMEs) of the total lipids were prepared using acid-catalyzed transesterification procedure and analyzed by gas chromatography–mass spectrometry (GC-MS) as described previously [28]. The measurements were performed on a PerkinElmer Clarus 600 T GC–MS (PerkinElmer, Inc., Shelton, CT, USA) in the following conditions: capillary column—Supelcowax 10 (60 m × 0.25 mm i.d., 0.25 μm film thickness; Supelco Inc., Bellefonte, PA, USA); carrier gas—helium (0.8 mL/min); the injection volume—0.5 μL (split ratio of 1:24); the injector temperature, 210 °C; the oven temperature was set at 140 °C, then ramped to 220 °C (7 °C/min), and held at 220 °C (23 min). The EI (positive ion electron impact) mass spectra were recorded at 70 eV using a trap current of 100 µA with an ion source temperature of 150 °C. The MS was scanned from *m/z* 22 to 395.

The identification of FA was accomplished by comparing their retention times with those of known standards (37 component FAME Mix, SUPELCO; Bellefonte, PA, USA) and the resulting mass spectra to those in the database (NIST MS Search 2.0). The amount of each FA was expressed as peak area percentage of total fatty acids.

### 2.5. Analysis of Volatile Compounds in Cheese

The extraction and further the separation and identification of volatile compounds from the cheese samples were achieved using in-tube extraction (ITEX) technique coupled with gas-chromatography hyphenated with mass spectrometry (GC-MS). For the extraction step, 3 g of each cheese sample were placed into a vial headspace and incubated at 60 °C for 30 min and under continuous agitation. The volatile compounds released in the headspace phase were adsorbed onto the ITEX fiber (ITEX-2TRAPTXTA, Tenax TA 80/100 mesh), which is part of the headspace syringe (CombiPAL AOC-5000 autosampler, CTC Analytics, Zwingen, Switzerland). The desorption of the volatile compounds was made into the GC-MS injector through thermal desorption.

The separation and identification of volatile compounds was carried out on a GC/MS QP-2010 (Shimadzu Scientific Instruments, Kyoto, Japan) model gas-chromatograph—mass, equipped with a Zebron ZB-5 ms capillary column of 30 m × 0.25 mm i.d and 0.25 mm film thickness. The column oven temperature program started at 40 °C (held for 10 min) and increased to 230 °C at a rate of 5 °C/min and was held at this end temperature for 5 min. The other main GC/MS parameters were as follows: the flow rate of the carrier gas (helium) was set at 1 mL/min, the split ratio was 1:2, the temperature of the injector and interface was set at 250 °C, the MS detector was operating in full scan, the scanned mass range was 40–650 *m*/*z*, the ionization energy was 70 eV. After the acquisition of the mass spectra, for the identification of the volatile compounds from cheese samples, they were compared with those of reference compounds from NIST27 and NIST147 mass spectra libraries. The amount of each volatile compound was quantified and expressed as percentages of the total peak area [29].

### 2.6. Statistical Analysis

All analyses were performed in duplicate, using MINITAB software. Differences were analyzed using one-way analysis of variance ANOVA (Analysis of Variance), general linear model, one-way ANOVA. Significance of differences (D1–D5) between means for each parameter was determined by Tukey’s test at a significance level of *p* < 0.05. All results were presented as mean ± SD (standard deviation).

## 3. Results and Discussion

### 3.1. Physico-Chemical Analyses

Table 1 shows the physico-chemical analyses performed on the five cheese samples during the 16 months of maturation. The moisture content of the samples, as expected, decreased from 54.66% to 46.69% during baking. These values were higher than those reported for Evora cheese [7], Valdeón cheese [11], and lower than those reported for Parmigiano Reggiano [5], Provolone del Monaco [30], Cheddar [31].

With the decrease of humidity, an increase in the amount of fat can be observed from 24.40% to 32.16%, respectively indicating a relative fat in the dry matter from 43.79% to 60.32%. In the same way, the amount of protein increases from 25.00% to 29.47. The values were close to those reported by D’Incecco et al. [5] or Brickley et al. [31]. The amount of mineral substances increased from 4.04% to 4.91%. Apuseni cheese ash is found in a much larger amount than that reported for Terrincho cheese [8] and in a slightly smaller amount than that reported for Prato cheese [32].

The salt concentration increased from 1.81% to 2.43%. This decrease is due to the moisture in the cheese during ripening. Similar values are reported for Cheddar cheese [31], Evora [7], or lower compared to Prato cheese [32] or Terrincho cheese [8].

The pH values detected in the analyzed samples were between 5.44 and 5.19, decreasing during ripening. During proteolysis, the formation of ionic groups occurs, which reduces the pH of the cheese and increases the acidity, and it is also related to the conversion of lactose into lactic acid by the action of bacteria present in yeast, using residual lactose [33]. These values were similar to those observed in other varieties, such as Cheddar [31,33], Terrincho [8], or Prato cheese [32].

### 3.2. Microbiological Analyses

The aim of the microbiological examination was monitoring the storage conditions (Table 2) and microbiological analyses for the cheese samples (Table 3) throughout the ripening period, which was in accordance with the requirements of the legislation in force.

Thus, for the microflora in the deposit, TYMC and TVC were monitored (Table 2), where an increase of TYMC from 0.31 cfu (colony-forming unit)/m^3^ × 10^2^ to 2.85 cfu/m^3^ × 10^2^ and of TVC from 2.05 cfu/m^3^ × 10^2^ to 5.82 cfu/m^3^ × 10^2^. These parameters are influenced by several factors such as temperature, humidity, ventilation, and personal activity. Even if these parameters have increased during the 16 months, they do not exceed the maximum allowed limits (TYMC max. 300 cfu/m^3^; TVC max. 600 cfu/m^3^) of the Order of the Ministry of Health−976/1998 [34].

For the Apuseni cheese samples (Table 3), the increase of TYMC from 0.12 cfu/g × 10^3^ to 16.52 cfu/g × 10^3^ during the 16 months of ripening is noticeable. A significant increase has been noticed in the case of TVC from 1.25 cfu/g × 10^2^ to 18.40 cfu/g × 10^2^. *Escherichia coli* was present only in D1—0 months of ripening, after which it is absent. The explanation for these results is that lactic acid bacteria are a heterogeneous group of microorganisms whose main activity is the degradation of carbohydrates with the formation of lactic acid, which, in the substrates in which it is synthesized, creates an unfavorable environment for the development of many species of pathogenic bacteria (*E. coli*, *Salmonella,* and *Shigella* species) [27], thus favoring the absence of *Salmonella. Staphylococcus aureus* increases from 0.90 cfu/m^3^ × 10^2^ to 7.80 cfu/m^3^ × 10^2^. According to regulation (ce) no. 2073/2005, the results for the cheese sample are within the maximum allowed limits of 1000 cfu/g. Last but not least, LAB increases from 0.55 cfu/m^3^ × 10^8^ to 0.90 cfu/m^3^ × 10^8^. These variants are due to the fact that *Staphylococcus aureus* is not inhibited by salt concentration.

Thus, during ripening cheese, it has a series of biochemical and microbiological changes. Microbiological changes during the 16 months include the continuation of process activities until the salt level in the moisture becomes inhibitory. Certain cheeses are characterized by the growth of microbiological parameters during ripening [35]. Even if the microbiological parameters increase (according to Table 3) during the 16 months of ripening, they fall within the maximum allowed limits.

### 3.3. Fatty Acid Composition

Most studies show the accumulation of fatty acids in cheese over long periods of maturation. The level of fatty acids gradually increases depending on the ripening time and the type of cheese [36,37,38,39].

The analysis of fatty acids in Apuseni cheese (Table 4) showed that most of the fatty acids were saturated, C16:0 (palmitic acid) being the most abundant. The amount of palmitic acid decreases from 38.43% in D1 to 35.60% after 16 months of maturation. The same C16:0 acid is predominant in Parmigiano Reggiano cheese, with an upward trajectory in a study conducted during 24 months of maturation [40]. The majority is found in Roncal, Mahon [41], Swiss cheese, Blue Cheese or Roquefort [42], Emmentaler [43], Ioannina, Arta, Evros, Thessaloniki, or Larissa [44]. Palmitic acid is followed by oleic acid (C18:1 *n*-9), which compared to palmitic acid increases with the maturation of the cheese from 19.86% to 28.11%, being the majority acid in Fontina Valle d’Aosta cheese [45], Cheddar [46], and Manchego [41]. Myristic acid (C14:0) is also found in large quantities, ranging from 13.62% to 12.22%. Similar values are reported for Emmentaler [43]. These fatty acids are followed by stearic acid (C18:0) with values between 7.57% and 9.57%; close values have been reported for Cheddar [46]. Dodecanoic (lauric) acid (C12:0) and decanoic (capric) acid (C10:0) have values between 7.57% and 9.57%. Butanoic (butyric) acid (C4:0) shows decreasing values during the ripening period, suggesting its selective release by lipases present in cheese or its synthesis by the cheese microflora [17,47,48]. The other fatty acids observed in Apuseni cheese are in a smaller amount, according to Table 4. The profile of fatty acids is specific to each type of cheese, this being one of the characteristics that makes them unique.

It is very important to note that the amount of saturated fatty acids decreases significantly (*p* < 0.05) with ripening from 77.93% to 68.42%, which are recognized as health risk factors, as well as polyunsaturated fatty acids decreasing from 1.72% to 1.11%. Among the PUFA are known components with anti-atherogenic action such as 18:2 *n*-6 belonging to class *n*-6 PUFA, and in the more important class *n*-3 PUFA, components such as 18:3 *n*-3, which are appreciated for their antithrombogenic effect [49].

Meanwhile, the amount of monounsaturated fatty acids increasing significantly (*p* < 0.05) from 20.82% to 28.98% was observed during the 16 months of baking the cheese. MUFAs have been very effective at reducing the risk of coronary heart disease. Indeed, MUFAs have been recognized as beneficial as the *n*-3 PUFA class for human health due to their lowering effect on blood cholesterol, especially DHA [49]. It is noteworthy that the ratio between *n*-6/*n*-3 decreases, which differed significantly (*p* < 0.05) from 8.55 in D1—0 months of maturation to 2.17 in D5—16 months of maturation. Similar results have been reported by Perotti et al. for Reggianito Argentino and Parmigiano Reggiano cheeses [50].

Animal nutrition has a major influence on the fatty acid profile of raw milk when ruminants have direct access to pasture, as is our case, increasing the presence of polyunsaturated fatty acids [51]. Cheese made from the milk of these cows is preferred by consumers [52]. Cheese is more than a source of essential nutrients (calcium, protein, vitamins); it contains many bioactive molecules, among which fatty acids are the most important. Over 400 distinct AFs have been detected in cheese fat, being considered the most complex fat present in the human diet [53].

In terms of fatty acid profile, Apuseni cheese has many features in common with Cheddar, Parmesan, and Emmentaler. Most of the fatty acids in Apuseni cheese (palmitic acid, oleic acid, myristic acid, stearic acid) are also predominant fatty acids in these types of cheese [38].

### 3.4. ITEX/GCeMS Profile of Volatile Compounds

The formation of taste and aroma substances that takes place from the D2—4 months of ripening for lactic acid gives the cheeses a sour, pleasant taste when it is in small quantities. The sour taste gradually disappears after the breakdown of lactic acid and is replaced by a taste similar to that of walnut kernels. The salt favors the individual highlighting of the different substances of taste and aroma but also the fat by the fact that it loosens the cheese paste and has an emulsifying effect. The fat-derived flavors associated with cheese ripening result from the release of fatty acids by lipolysis and further modification of fatty acids by microorganisms to other compounds. Protein hydrolysis products influence the taste and aroma of cheeses all the more intensely as a further decomposition has taken place. In the case of strongly ripening cheeses, as in the case of Apuseni cheese, the presence of amines and ammonia contributes to the formation of the characteristic taste and aroma.

A total of 24 volatiles were found of which 22 were tentatively identified based on their mass spectra and retention indices from spectra databases and published data (Table 5). The volatile constituents present in the Apuseni cheese samples include alcohols, aldehydes, ketones, terpenoids, esters, as well as other classes of compounds. The most abundant group in all the sample cheeses was that of ketones. The ketones group was also found to be the major group of La Serena cheese [54], which is a cream cheese [46]. The majority compound in this class is 2-heptanone, which is present starting with 8 months of ripening and reaching 54.97% after 16 months of maturation. 2-Heptanone is the major compound reported in other cheeses such as Maroilles, Pont l’Eveque, Langres, Vacherin, Livarot, Limburger, Beaufort, Appenzeller, Chaumes, and Raclette [55]. In addition, from the ketones group, acetophenones are found in a large amount—68.66% in D1, and they decrease to 4.90% in D5. Acetophenones are also part of the majority of cheeses: Maroilles, Livarot, Pont l’Eveque, Langres, and Limburger [55].

Aldehydes, the majority group for Bitto cheese, is represented in Apuseni cheese by benzaldehyde, which decreases from 10.52% in D1 to 2.57% in D5. This compound is predominantly for Comte, Raclette, Tete de Moine, Livarot, and Tilsit [55].

Several compounds from the terpene group were identified, but they were not identified at each stage of ripening (Table 5). The compounds that were identified in a larger amount, starting with D2, are limonene and thymol. This class of volatile compounds is detected mainly in artisanal cheeses made in the Alpine regions. These compounds are primarily derived from differences in the feeding regime of cows [56]. As described above, the milk used for Apuseni cheese comes from small producers in the Apuseni Mountains. It is possible that the terpene content of the cheese produced in the alpine region is high due to the diverse wild flora on which the cows graze [57]. However, a high terpene content has been reported for Montasio cheese during ripening, which is due to the properties of lactic acid bacteria to modify and biosynthesize terpenoids [58].

Regarding the alcohols class, they are present in a very small amount, only in D2—4 months of ripening and only 2,3-butanediol and phenol. The presence of phenolic compounds in milk is related to the process of conjugation, which is a detoxification mechanism that enables an animal to solubilize xenobiotics and to excrete them, usually in urine. It seems that the abundance of phenolic compounds in sheep milk is influenced by both feeding and breed [59]. These compounds are also reported in Parmigiano Reggiano [55] or Turkish white cheese [60].

Esters are valued for their ability to provide sweet, fruity, or floral notes but also for minimizing the sharp and bitter taste of the cheese, which is usually associated with high levels of free fatty acids (FFA) and amines [55]. The major esters of Apuseni cheese are Butanoic acid, ethyl ester, and hexanoic acid, ethyl ester. Predominant quantities were also reported for Parmigiano Reggiano, Limburger, Chaumes, Tilsit, and Turkish white cheese [4,5,53,55,60].

All these biochemical processes that take place during the maturation of the cheese modify the physical and chemical parameters of the Apuseni cheese and include three main reactions: metabolism of the residual lactose, proteolysis, and lipolysis. The enzymes involved in the ripening process can come from several sources: the lipase lipoproteins in milk survive after pasteurization and participate in lipolysis. In addition to their main fermentation function, LAB provides enzymes, including proteinase and esterase. LAB are the main factors involved in the maturation process of the cheese due to its strong metabolic activity. Lactic bacteria metabolize lactose, the main carbohydrate in milk, into lactic acid, after fermentation, through a process called glycolysis. This process is responsible for acidifying the cheese. Proteolysis is the process that deals with the metabolism of various caseins, in smaller peptides and free amino acids by endogenous proteins in milk, and other proteolytic enzymes produced by LAB. Proteinases and peptidases catalyze the cleavage of the polypeptides chains to produce free amino acids, which undergo several biochemical reactions that result in flavor compounds. The ripening time influences the degree of proteolysis. The longer the storage time, the higher the degree of proteolysis. Lipolysis is the catabolic process leading to the breakdown of triacylglycerols (TAGs) into FFAs and glycerol. The released FFAs contribute directly to the cheese flavor by imparting specific fatty acid flavor notes or indirectly as precursors for the formation of other flavor compounds [61,62].

## 4. Conclusions

In recent years, the consumption of ripening cheese and premium products has increased. Apuseni cheese is a premium, ripening cheese, obtained from milk from the Apuseni mountains, which gives it uniqueness. Given the long ripening time, the price for this cheese is higher; therefore, it was important to distinguish the changes that take place during the ripening process. It is a cheese that is sold in different stages of ripening. Humidity is the component that changes the most and influences the physicochemical components. The microbiological activity is not negatively influenced during the 16 months. The profiles of fatty acids and volatile compounds are closely related and have shown continuous changes along the ripening path. In conclusion, this study showed that the duration of ripening deserves to be highlighted to demonstrate that with ripening, Apuseni cheese increases its nutritional values, the costs being justified for a premium product.

## Figures and Tables

**Figure 1 foods-10-00258-f001:**
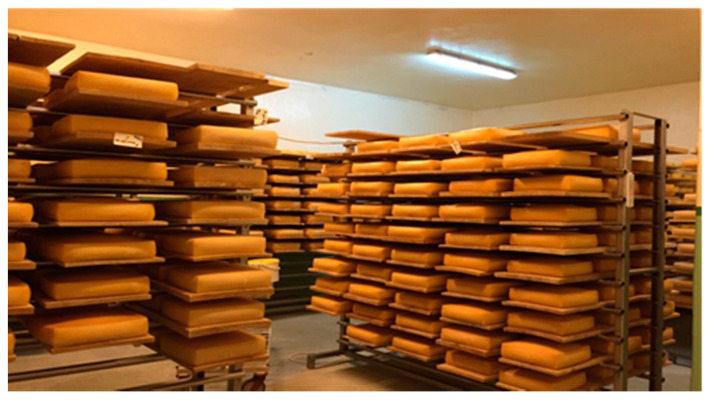
Apuseni cheese.

**Figure 2 foods-10-00258-f002:**
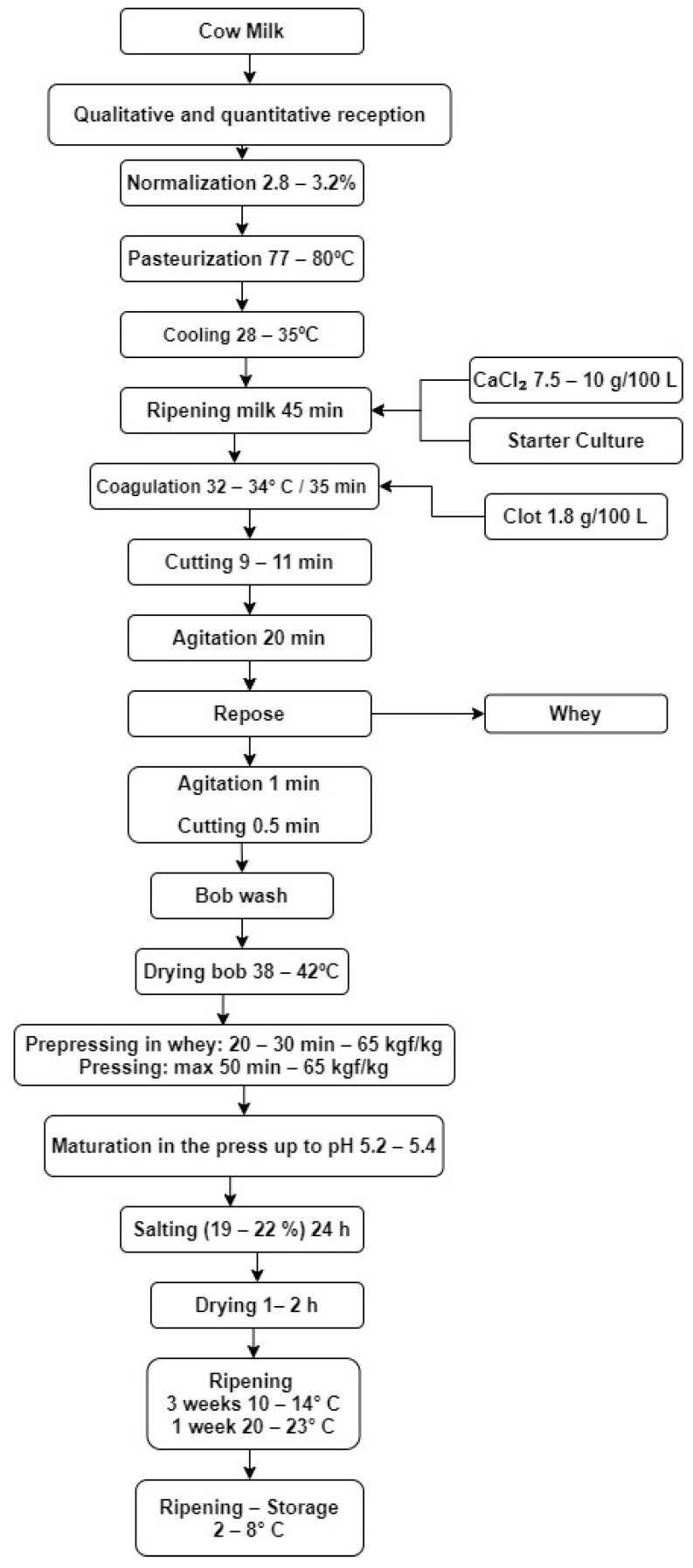
Flow diagram describing the technological steps of the Apuseni cheese.

**Table 1 foods-10-00258-t001:** Physico-chemical properties of cheese samples during ripening.

Samples	Moisture (%)	Fat (%)	Fat in Dry Matter (%)	Ash (%)	Total Protein (%)	NaCl (%)	pH
D1	54.66 ± 0.49 ^a^	24.40 ± 0.23 ^d^	43.79 ± 0.28 ^d^	4.04 ± 0.12 ^c^	25.00 ± 0.17 ^d^	1.81 ± 0.08 ^d^	5.44 ± 0.02 ^a^
D2	52.91 ± 0.36 ^a^	27.55 ± 0.48 ^c^	57.55 ± 0.45 ^c^	4.32 ± 0.10 ^bc^	26.71 ± 0.36 ^c^	1.93 ± 0.04 ^cd^	5.37 ± 0.03 ^ab^
D3	50.04 ± 0.59 ^b^	29.11 ± 0.33 ^b^	58.46 ± 0.46 ^bc^	4.56 ± 0.09 ^ab^	27.25 ± 0.20 ^bc^	2.12 ± 0.05 ^bc^	5.32 ± 0.03 ^bc^
D4	47.80 ± 0.40 ^c^	31.56 ± 0.19 ^a^	59.77 ± 0.28 ^ab^	4.74 ± 0.07 ^a^	28.21 ± 0.32 ^ab^	2.30 ± 0.06 ^ab^	5.23 ± 0.02 ^cd^
D5	46.69 ± 0.39 ^c^	32.16 ± 0.23 ^a^	60.32 ± 0.27 ^a^	4.91 ± 0.06 ^a^	29.47 ± 0.54 ^a^	2.43 ± 0.06 ^a^	5.19 ± 0.02 ^d^

Identical superscripts letters within rows indicate no significant difference (*p* > 0.05); D1—0 months of ripening; D2—4 months of ripening; D3—8 months of ripening; D4—12 months of ripening; D5—16 months of ripening.

**Table 2 foods-10-00258-t002:** Microbiological parameters in the storage space during cheese ripening.

Samples	D1	D2	D3	D4	D5
TYMC (cfu/m^3^ × 10^2^)	0.31 ± 0.03 ^e^	0.83 ± 0.02 ^d^	1.55 ± 0.06 ^c^	2.45 ± 0.07 ^b^	2.85 ± 0.08 ^a^
TVC (cfu/m^3^ × 10^2^)	2.05 ± 0.06 ^c^	2.10 ± 0.11 ^c^	3.40 ± 0.11 ^b^	5.50 ± 0.07 ^a^	5.82 ± 0.04 ^a^

Identical superscripts letters within rows indicate no significant difference (*p* > 0.05); D1—0 months of ripening; D2—4 months of ripening; D3—8 months of ripening; D4—12 months of ripening; D5—16 months of ripening.

**Table 3 foods-10-00258-t003:** Microbiological parameters of cheese samples during ripening.

Samples	TYMC (cfu/g × 10^3^)	TVC (cfu/g × 10^2^)	*Escherichia coli* (cfu/g × 10^2^)	*Staphylococcus aureus* (cfu/g × 10^2^)	*Salmonella* (cfu/25 g)	LAB (cfu/g × 10^8^)
D1	0.12 ± 0.01 ^e^	1.25 ± 0.06 ^e^	5.02 ± 0.05	0.90 ± 0.03 ^d^	n.d.	0.55 ± 0.07 ^a^
D2	2.68 ± 0.08 ^d^	4.50 ± 0.14 ^d^	not detected	1.07 ± 0.06 ^d^	n.d.	0.64 ± 0.08 ^a^
D3	8.70 ± 0.13 ^c^	6.20 ± 0.13 ^c^	not detected	2.35 ± 0.15 ^c^	n.d.	0.73 ± 0.18 ^a^
D4	12.4 ± 0.14 ^a^	10.50 ± 0.08 ^b^	not detected	4.37 ± 0.22 ^b^	n.d.	0.84 ± 0.17 ^a^
D5	16.52 ± 0.11 ^a^	18.40 ± 0.07 ^a^	not detected	7.80 ± 0.14 ^a^	n.d.	0.90 ± 0.13 ^a^

Identical superscripts letters within rows indicate no significant difference (*p* > 0.05); D1—0 months of ripening; D2—4 months of ripening; D3—8 months of ripening; D4—12 months of ripening; D5—16 months of ripening; n.d. = not detected.

**Table 4 foods-10-00258-t004:** Fatty acid composition (%) of Apuseni cheese samples.

No	Fatty Acids (% of Total Fatty Acids)	D1	D2	D3	D4	D5
1	C4:0	3.035 ± 0.021 ^a^	2.680 ± 0.014 ^b^	2.580 ± 0.028 ^c^	2.200 ± 0.014 ^d^	1.555 ± 0.021 ^e^
2	C6:0	3.270 ± 0.014 ^a^	2.795 ± 0.021 ^b^	2.695 ± 0.021 ^c^	2.320 ± 0.028 ^d^	2.085 ± 0.021 ^e^
3	C8:0	1.689 ± 0.001 ^c^	1.700 ± 0.001 ^b^	1.712 ± 0.001 ^a^	1.722 ± 0.002 ^a^	1.117 ± 0.005 ^d^
4	C10:0	3.380 ± 0.001 ^d^	3.780 ± 0.001 ^b^	3.812 ± 0.002 ^a^	2.931 ± 0.002 ^e^	3.510 ± 0.002 ^c^
5	C12:0	3.660 ± 0.028 ^b^	3.360 ± 0.042 ^c^	4.200 ± 0.014 ^a^	3.685 ± 0.021 ^b^	3.760 ± 0.014 ^b^
6	C14:0	13.615 ± 0.021 ^c^	13.640 ± 0.042 ^c^	13.870 ± 0.028 ^b^	14.090 ± 0.071 ^a^	12.215 ± 0.021 ^d^
7	C15:0	1.281 ± 0.004 ^a^	1.140 ± 0.001 ^b^	1.101 ± 0.002 ^c^	1.000 ± 0.001 ^d^	0.920 ± 0.003 ^e^
8	C16:0	38.425 ± 0.049 ^a^	38.250 ± 0.028 ^a^	36.985 ± 0.078 ^b^	36.670 ± 0.057 ^c^	35.603 ± 0.053 ^d^
9	C16:1 *n*-9	0.901 ± 0.001 ^a^	0.681 ± 0.003 ^b^	0.420 ± 0.001 ^c^	0.260 ± 0.001 ^d^	0.250 ± 0.001 ^e^
10	C18:0	9.570 ± 0.028 ^a^	7.940 ± 0.035 ^c^	7.567 ± 0.045 ^d^	8.195 ± 0.021 ^b^	7.640 ± 0.049 ^d^
11	C18:1 *n*-9	19.860 ± 0.014 ^e^	22.390 ± 0.085 ^d^	23.718 ± 0.067 ^c^	24.588 ± 0.039 ^b^	28.113 ± 0.032 ^a^
12	C18:1 *n*-7	0.059 ± 0.001 ^e^	0.130 ± 0.004 ^d^	0.272 ± 0.009 ^c^	0.500 ± 0.004 ^b^	0.620 ± 0.003 ^a^
13	C18:2 *n*-6	1.540 ± 0.001 ^a^	1.260 ± 0.002 ^b^	0.950 ± 0.001 ^c^	0.870 ± 0.001 ^d^	0.761 ± 0.002 ^e^
14	C18:3 *n*-3	0.181 ± 0.005 ^c^	0.300 ± 0.007 ^b^	0.311 ± 0.001 ^b^	0.341 ± 0.008 ^a^	0.351 ± 0.004 ^a^
ƩSFAs		77.930 ± 0.028 ^a^	75.290 ± 0.014 ^b^	74.515 ± 0.049 ^c^	72.828 ± 0.025 ^d^	68.423 ± 0.018 ^e^
ƩMUFAs		20.820 ± 0.014 ^e^	23.200 ± 0.014 ^d^	24.415 ± 0.021 ^c^	25.340 ± 0.014 ^b^	28.975 ± 0.021 ^a^
ƩPUFAs		1.722 ± 0.006 ^a^	1.560 ± 0.014 ^b^	1.259 ± 0.005 ^c^	1.210 ± 0.014 ^d^	1.109 ± 0.005 ^e^
*n*-6/*n*-3		8.545 ± 0.021 ^a^	4.203 ± 0.018 ^b^	3.060 ± 0.014 ^c^	2.550 ± 0.014 ^d^	2.170 ± 0.042 ^e^

Identical superscripts letters within rows indicate no significant difference (*p* > 0.05); D1—0 months of ripening; D2—4 months of ripening; D3—8 months of ripening; D4—12 months of ripening; D5—16 months of ripening. Values (mean ± SD, *n* = 3) SFAs—saturated fatty acids, MUFAs—monounsaturated fatty acids, PUFAs—polyunsaturated fatty acids, Butanoic (butyric) acid (C4:0); Hexanoic (caproic) acid (C6:0); Octanoic (caprylic) acid (C 8:0); Decanoic (capric) acid (C10:0); Dodecanoic (lauric) acid (C12:0); Myristic acid (C14:0); Pentadecanoic acid (C15:0); Palmitic acid (C16:0); Hexadecenoic acid (C16:1 *n*-9); Stearic acid (C18:0); Oleic acid (C18:1 *n*-9); Vaccenic acid (C18:1 *n*-7); Linoleic acid (C18:2 *n*-6); Linolenic acid (C18:3 *n*-3).

**Table 5 foods-10-00258-t005:** Volatile compounds (%) identified in Apuseni cheese samples at five different stages of ripening.

Compound Name	RI (Retention Indices)	D1	D2	D3	D4	D5	Odor Characteristic Descriptors
**Alcohols**							
2,3-Butanediol	771	n.d.	0.96 ± 0.03	n.d.	n.d.	n.d.	fruity
**Phenol**	981	n.d.	3.85 ± 0.07	n.d.	n.d.	n.d.	phenolic
Aldehydes							
Benzaldehyde	958	10.52 ± 0.04 ^a^	1.41 ± 0.02 ^e^	1.97 ± 0.04 ^d^	2.84 ± 0.02 ^b^	2.57 ± 0.05 ^c^	almond, burnt sugar
**Ketones**							
2-Heptanone	896	n.d.	11.99 ± 0.19 ^c^	24.53 ± 0.11 ^b^	12.25 ± 0.11 ^c^	54.97 ± 0.26 ^a^	sulfur, pungent, green, fruity
Acetophenone	1042	68.66 ± 0.39 ^a^	8.78 ± 0.12 ^c^	10.52 ± 0.10 ^b^	7.88 ± 0.16 ^d^	4.90 ± 0.13 ^e^	sweet, flower, almond
2-Nonanone	1094	n.d.	2.96 ± 0.11 ^b^	n.d.	n.d.	5.20 ± 0.21 ^a^	fruity, floral
Benzophenone	1605	n.d.	n.d.	1.18 ± 0.10 ^b^	1.97 ± 0.08 ^a^	1.45 ± 0.16 ^b^	faint, sweet rose, light herbal
Ethanone, 1-(4-methylphenyl)-	1182	n.d.	7.14 ± 0.18	n.d.	n.d.	n.d.	almond, sweet, floral, hay
**Terpene hydrocarbons and oxygenated derivatives**							
p-Cymene	1028	n.d.	1.37 ± 0.05	n.d.	n.d.	n.d.	citrus
Caryophyllene	1468	n.d.	3.22 ± 0.18	n.d.	n.d.	n.d.	woody, spicy, fruity, sweet
α-Caryophyllene	1454	n.d.	3.95 ± 0.14	n.d.	n.d.	n.d.	fruity, woody
Limonene	1031	n.d.	10.8 ± 0.16 ^b^	8.46 ± 0.09 ^c^	12.90 ± 0.13 ^a^	6.26 ± 0.21 ^b^	citrus, mint
2,6-Octadienal, 3,7-dimethyl-, (E)-	1255	20.82 ± 0.23	n.d.	n.d.	n.d.	n.d.	lemon
Verbenone	1205	n.d.	1.76 ± 0.10 ^a^	1.72 ± 0.09 ^a^	n.d.	n.d.	spicy
Thymol	1291	n.d.	22.53 ± 0.37 ^a^	20.42 ± 0.26 ^b^	9.16 ± 0.19 ^c^	3.66 ± 0.09 ^d^	herbal, thyme, earthy
Carvacrol	1299	n.d.	2.46 ± 0.35 ^a^	3.23 ± 0.11 ^a^	0.84 ± 0.07 ^b^	n.d.	caraway
**Esters**							
Butanoic acid, ethyl ester	805	n.d.	11.60 ± 0.13 ^c^	13.54 ± 0.21 ^c^	23.92 ± 0.13 ^a^	4.91 ± 0.04 ^d^	fruity, sweet
Acetic acid, butyl ester	813	n.d.	n.d.	n.d.	14.92 ± 0.16 ^a^	4.13 ± 0.11 ^b^	fruity, sweet, green, pear-like
Butanoic acid, propyl ester	897	n.d.	n.d.	n.d.	4.57 ± 0.13 ^a^	2.60 ± 0.13 ^b^	pineapple, solvent
Butanoic acid, butyl ester	993	n.d.	n.d.	n.d.	n.d.	1.21 ± 0.16	fresh, sweet, fruity
Hexanoic acid, ethyl ester	1003	n.d.	1.54 ± 0.22 ^c^	3.89 ± 0.14 ^ab^	4.46 ± 0.14 ^a^	3.36 ± 0.11 ^b^	fruity, fresh, sweet
**Others**							
n.i.		n.d.	1.55 ± 0.08	n.d.	n.d.	n.d.	
Benzoic Acid	1174	n.d.	n.d.	10.54 ± 0.22 ^a^	4.27 ± 0.18 ^b^	4.79 ± 0.28 ^b^	winey, balsamic
n.i.		n.d.	2.13 ± 0.17 ^a^	1.72 ± 0.13 ^a^	n.d.	n.d.	

Identical superscripts letters within rows indicate no significant difference (*p* > 0.05); D1—0 months of ripening; D2—4 months of ripening; D3—8 months of ripening; D4—12 months of ripening; D5—16 months of ripening; n.i. = not identified; n.d. = not detected.

## Data Availability

The primary source data are not publicly available due to the institutional data policy. However, the data from this study are available upon request to the corresponding author.

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
