# Peer review of "Changes in Physicochemical and Microbiological Properties, Fatty Acid and Volatile Compound Profiles of Apuseni Cheese during Ripening"

_foods, 2021, doi:10.3390/foods10020258_

Round 1
Reviewer 1 Report
The Authors reported an interesting work on the changes in the physichchemical and microbiological properties of Apuseni cheese during ripening. Manuscript cannot be accepted for publication, but requires a major revision.
Abstract
It is a well-known fact that during cheese ripening the moisture decreases and the dry matter increases. But there is no information how the ripening time affected the profile of volatile compounds, which is much more interesting.
Keywords
Insert key-words different from the words already used in the title.
Introduction
The authors should include information about Apuseni cheese in the Introduction. What kind of cheese is it? What is its origin, its unique characteristics?
Materials and methods
This chapter is the weakest point of the work. The methodological assumptions are incomprehensible. How was the cheese selected for testing? The description of the methodology shows that these were random cheeses from a dairy. How many cheeses exactly were tested? Did the cheeses S1 - S5 come from the same production batch or from others?
What milk was used to make these cheese? (species, breed, feed)
How long did the cheeses ripen - 15 or 16 months? L82 vs. L86
L87 How was milk standardised? By fat content? To what level?
L88 -89 Add more details about additives in cheese production; what quantities of cultures, enzyme were used, what was the strength of the coagulating enzyme? manufacturer's data of additives.
what was the weight of the cheese wheels?
how were the cheeses salted?
what was the humidity during ripening?
Was the surface of the cheeses in any way protected? How were the cheese wheels cared for during ripening?
L95 analyses were carried out every three or four months?
In how many repetitions were the analyses carried out?
L174 How long have the volatile compounds been desorbed from the fiber into the injector?
Authors should add the LRIs of the determined volatile compounds to Table 5.
L190 Explain what it means: "The results of three independent assays (performed with replicates each". What statistical programme was used for the analysis?
L204 Was this difference really not statistically significant? use "moisture" instead of "humidity".
Add description about mean values and standard deviations to tables 1-3.
L223 This presentation of the significance of the differences is not understandable. It would be much better to use, for example " the averages given with different letters in the same row indicate significant differences at the 0.05 level from each other".
The criterion for attributing the letters, relating to significance, must always be the same, otherwise confusion is created. the letter a must always be attributed to the smallest mean and the letters b, c and d gradually in ascending order.
L233 What are the permitted maximum values?
L238 Check the units, cfu/m3?
L289 There were no statistically significant changes in the profile of fatty acids?
L313 The way the results are discussed should be improved. The authors discuss the individual volatile compounds present in S1-S5 samples, but how did the ripening of the cheese affect the profile of the volatile compounds?
L349 What about standard deviations values? statistically significant differences?
Conclusions
The authors state " In addition, the results can serve as a basis for the 359 development of comparative analysis tools and strategies aimed at improving the nutritional 360 characteristics of cheese maturation. Also, studies of the volatile profile of cheese could be an interesting 361 method for studying the incorporation of further advantages such as the inclusion of certain commercial 362 starters or the application of other treatments to enhance the quality or the shelf-life of cheeses." In what way? The authors state that Apusani cheese is sold in different ripening stages. There is no answer to the question of which period of ripening the cheese has the best nutritional value? This would be very valuable information for both producers and consumers.
In order to better assess the profile of fatty acids, authors should give values of thrombogenic and atherogenic indexes of cheeses in particular ripening stages.
There is no sensorial analysis of the cheeses that would enrich the work and support its commercial development. The Authors of the paper repeatedly highlighted the importance of sensory characteristics for consumers. Information on how changes in the profile of volatile compounds and fatty acids during cheese ripening have influenced their sensory characteristics is crucial for the acceptability of the product among consumers.
Author Response
Ms. Ref. No.: Foods- 1057596
Title: Changes in physicochemical and microbiological properties, fatty acid and volatile compound profiles of Apuseni cheese during ripening
Foods –MDPI – Section “Dairy” Special Issue "Functional and Nutritional Properties of Different Kinds of Milk"
On behalf of all co-authors,
I would like to thank you for taking necessary steps to improve our Manuscript in term of scientific quality. I also thank all the reviewers for spending their valuable time to correct our MS entitled “CHANGES IN PHYSICOCHEMICAL AND MICROBIOLOGICAL PROPERTIES, FATTY ACID AND VOLATILE COMPOUND PROFILES OF APUSENI CHEESE DURING RIPENING”. Based on your and the Reviewers’ valuable comments and suggestions, we revised our MS and we hope, the changes made in the revised MS are satisfactory.
All changes are marked as highlighted red letters in the revised MS.
Thank you for your confidence – according to your suggestions, we had performed amendments to our MS in order to better highlight its scientific potential.
Reviewers' comments:
The Authors reported an interesting work on the changes in the physichchemical and microbiological properties of Apuseni cheese during ripening. Manuscript cannot be accepted for publication, but requires a major revision.
Abstract
It is a well-known fact that during cheese ripening the moisture decreases and the dry matter increases. But there is no information how the ripening time affected the profile of volatile compounds, which is much more interesting.
Response:
Abstract: Cheese is one of the oldest foods, being known over 4000 types, appreciated for its nutritional, sensory and high quality fatty acid properties. Cheese ripening is basically about the breakdown of proteins, lipids and carbohydrates (acids and sugars) which releases flavour compounds and modifies cheese texture. The formation of taste and aroma substances takes place from the S2 phase of ripening: lactic acid gives the cheeses a sour, pleasant taste when it is in small quantities. The sour taste gradually disappears after the breakdown of lactic acid and is replaced by a taste similar to that of walnut kernels. The salt favors the individual highlighting of the different substances of taste and aroma, but also the fat that loosens the cheese paste and has an emulsifying effect. The fat derived flavors associated with cheese ripening result from the release of fatty acids by lipolysis and further modification of fatty acids by microorganisms to other compounds. Protein hydrolysis products influence the taste and aroma of cheeses, all the more intensely as a further decomposition has taken place. In the case of strongly ripening cheeses, as in the case of Apuseni Cheese, the presence of amines and ammonia contributes to the formation of the characteristic taste and aroma. The purpose of this study is to analyze the influence of ripening time (16 months) on the chemical properties, fatty acids and volatile componds of Apuseni cheese to improve knowledge about the quality of ripening cheese. As the cheese matures, it notices a decrease in sample moisture and pH, at the same time an increase in the content of fat, protein, ash and NaCl. Even if the microbiological parameters increase during the 16 months of ripening, they fall within the maximum allowed limits.During ripening, 14 fatty acids were identified, with a decrease in saturated fatty acids from 77.93% to 68.42% and an increase in monounsaturated fatty acids from 20.82% to 28.98. The volatile compounds identified in the cheese samples are 2 alcohols, 1 aldehyde, 5 ketones, 8 hydrocarbon terpenes and oxygenated derivatives, as well as 5 esters and 3 other classes of compounds.
Keywords
Insert key-words different from the words already used in the title.
Response:
matured cheese, nutrients, lipids, aroma compounds
Introduction
The authors should include information about Apuseni cheese in the Introduction. What kind of cheese is it? What is its origin, its unique characteristics?
Response:
Cheese ripening is basically about the breakdown of proteins, lipids and carbohydrates (acids and sugars) which releases flavour compounds and modifies cheese texture. In the broadest terms there are three sources of cheese flavour:
Flavours present in the original cheese milk, such as natural butter fat flavour and feed flavour. Breakdown products of milk proteins, fats and sugars which are released by microbial enzymes, enzymes endogenous to milk, and enzyme additives. Flavour and texture development are strongly dependent on: pH profile, composition, salting, temperature, humidity, experience.
As a general rule factors which increase the rate of ripening increase the risk of off flavour development, and reduce the period of time when the cheese is saleable. Protein degradation during cheese curing is a directed process resulting in protein fragments with desirable flavours. Dairy fat is a wonderfully rich source of flavours, because it contains an extremely diverse selection of fatty acids.
Dairy fat without any ripening during cheese making is an important contributor to cheese flavour and texture. Fat also acts as a flavour reservoir, so hydrophobic (fat soluble) flavours derived from protein breakdown are stored in the fat and released during mastication in the mouth. The fat derived flavours associated with cheese ripening result from the release of fatty acids by lipolysis and further modification of fatty acids by microorganisms to other compounds.
Benzaldehyde and (E,E)-2,4-nonadienal are unsaturated aldehydes formed by oxidation of unsaturated fatty acids. Instead, acetaldehyde is formed via lactic acid fermentation. These compounds are formed during the secondary phase of the auto-oxidation process and give rise to off-flavors.
Apuseni cheese is a hard cheese, originally from Coltesti- Apuseni, Romania, it is a royal delicacy balanced delicately between sweet and salty. It is recognized by the holes formed in the decomposition process of lactic acid and glutamic acid, when existing microorganisms produce carbon dioxide that remains trapped in the dense mass of cheese. The taste is specific, slightly sweet and is due to propionic ferments.
Materials and methods
This chapter is the weakest point of the work. The methodological assumptions are incomprehensible. How was the cheese selected for testing? The description of the methodology shows that these were random cheeses from a dairy. How many cheeses exactly were tested? Did the cheeses S1 - S5 come from the same production batch or from others?
What milk was used to make these cheese? (species, breed, feed)
L87 How was milk standardised? By fat content? To what level?
L88 -89 Add more details about additives in cheese production; what quantities of cultures, enzyme were used, what was the strength of the coagulating enzyme? manufacturer's data of additives.
what was the weight of the cheese wheels?
how were the cheeses salted?
what was the humidity during ripening?
Was the surface of the cheeses in any way protected? How were the cheese wheels cared for during ripening?
Response:
The samples of ‘Apuseni’ cheeses were taken from a milk processing unit in Romania (Coltesti dairy products).
The ripening of the samples took place over a period of 16 months. Sampling has been carried out over several periods, even in different seasons, in order to have continuity and to be able to tell the true quality of this type of hard paste cheese: the first sampling: S1- 0 months of ripening; second sampling: S2- 4 months of ripening; third sampling: S3- 8 months of ripening; fourth sampling: S4- 12 months of ripening; fifth sampling: S5- 16 months of ripening.
Cheesemaking: The supply of cow's milk is made exclusively from authorized farms within a radius of 30 kilometers from the Colţeşti-Alba-Romania area, which supply the best quality milk, according to European norms. The normalization of milk is done at 2.8 - 3.2% g. For hard cheeses, the acidity of matured milk does not exceed 18 - 20ºT. Standardized cow's milk was pasteurized at 65° C for 30 min.
The cultures used are frozen Cryofast cultures, coagulation power 1800 IMCU/g, produced by Company of Sacco System, Italy. The lyophilized culture Lactococcus lactis ssp. Lactis, Streptococcus thermophiles and CaCl2 were added, and in the cogulation stage at 32-34° C / for 35 min the microbial enzymatic clot (Rhizomucor miehei), produced by Ideal Still Exim Srl, 1.8 g / 100 l, for 30 - 35 minutes was added. This was followed by the cutting of the curd, the separation of whey, a slight heating to 40° C, mixing and a new separation of whey with the formation of the curd. Curds formed, pressed case was subjected to salting, drying and maturation ripening in storage at temperatures of 13-14°C / for 15 days and then 2-8°C for 15 16 months.
The shape of the blocks is parallelepiped, weight 7 kg, height 7-10 cm, length 25 cm, width 25 cm.
The salting process is wet, in saturated brine, and the pH is 5.02. Typically, the temperature of the brine and the air in the salting room varies between 8 and 16 ° C, the relative humidity of the air being 90-95%. The duration of salting is 24 hours, after which it takes 1-2 hours.
The ripening cheese is kept at a temperature of 2 - 8 ° C and a relative humidity of 85 - 90%. Periodically, the shell is washed with salt water, and if the shell is too thick, it is scraped. The turning prevents the cheese from deforming, ensuring a uniform maturation throughout the mass. The cheeses are turned by hand. Samples were taken from the same batch (batch 90), in all phases S1-S5. The sample was taken from the same blocks, with a scabbard.
How long did the cheeses ripen - 15 or 16 months? L82 vs. L86
Response:
Corrected: 16
L95 analyses were carried out every three or four months?
Response:
Corrected: four
In how many repetitions were the analyses carried out?
L174 How long have the volatile compounds been desorbed from the fiber into the injector?
Response: The parameters that can be set for the volatile compounds extraction using the ITEX technique and the CombiPal autosampler software don't include the desorption time. Instead it included the desorption speed and it was set to 50 uL/s
Authors should add the LRIs of the determined volatile compounds to Table 5.
Response: the LRIs were added in Table 5.
L190 Explain what it means: "The results of three independent assays (performed with replicates each". What statistical programme was used for the analysis?
Response:
For each test, 2 repetitions were performedwere expressed as mean value ± SD and analysed by means of the three-way ANOVA (Analysis of Variance) General Linear Model, one-way ANOVA and Tukey’s comparison statistical tests; for each parameter Tukey's comparison tests being performed at a 95% confidence level.
L204 Was this difference really not statistically significant? use "moisture" instead of "humidity".
Response:
Corrected: The moisture of the samples, as expected, decreased from 54.66% to 46.69% during ripening, statistically insignificant (p< 0.05), but statistically insignificant (p> 0.05) for the first sample S1-S2 and last samples S4-S5.
L223 This presentation of the significance of the differences is not understandable. It would be much better to use, for example " the averages given with different letters in the same row indicate significant differences at the 0.05 level from each other".
The criterion for attributing the letters, relating to significance, must always be the same, otherwise confusion is created. the letter a must always be attributed to the smallest mean and the letters b, c and d gradually in ascending order.
Response:
I corrected, but in Minitab a is the highest value, and b, c, d decrease in descending order.
Proof picture below:
L233 What are the permitted maximum values?
Response:
Corrected: TYMC max. 300 cfu/m3; TVC max. 600 cfu/m3
L238 Check the units, cfu/m3?
Response:
Corrected: For the Apuseni cheese samples (Table 3), the increase of TYMC from 0.12 cfu / g* 103 to 16.52 cfu / g * 103 during the 16 months of ripening is noticeable. A significant increase being noticed in the case of TVC from 1.25 cfu / g * 102 to 18.40 cfu / g* 102
L289 There were no statistically significant changes in the profile of fatty acids?
Response:
Corrected: The results of fatty acids are interpreted statistically. You can find them attached in the manuscript Table 4
L313 The way the results are discussed should be improved. The authors discuss the individual volatile compounds present in S1-S5 samples, but how did the ripening of the cheese affect the profile of the volatile compounds?
Response:
The formation of taste and aroma substances takes place from the S2 phase of ripening: lactic acid gives the cheeses a sour, pleasant taste when it is in small quantities. The sour taste gradually disappears after the breakdown of lactic acid and is replaced by a taste similar to that of walnut kernels. The salt favors the individual highlighting of the different substances of taste and aroma, but also the fat by the fact that it loosens the cheese paste and has an emulsifying effect. The fat derived flavors associated with cheese ripening result from the release of fatty acids by lipolysis and further modification of fatty acids by microorganisms to other compounds. Protein hydrolysis products influence the taste and aroma of cheeses, all the more intensely as a further decomposition has taken place. In the case of strongly ripening cheeses, as in the case of Apuseni Cheese, the presence of amines and ammonia contributes to the formation of the characteristic taste and aroma.
L349 What about standard deviations values? statistically significant differences?
Response:
Corrected: we have no standard deviation for volatile compounds
Conclusions
The authors state " In addition, the results can serve as a basis for the 359 development of comparative analysis tools and strategies aimed at improving the nutritional 360 characteristics of cheese maturation. Also, studies of the volatile profile of cheese could be an interesting 361 method for studying the incorporation of further advantages such as the inclusion of certain commercial 362 starters or the application of other treatments to enhance the quality or the shelf-life of cheeses." In what way? The authors state that Apusani cheese is sold in different ripening stages. There is no answer to the question of which period of ripening the cheese has the best nutritional value? This would be very valuable information for both producers and consumers.
In order to better assess the profile of fatty acids, authors should give values of thrombogenic and atherogenic indexes of cheeses in particular ripening stages.
There is no sensorial analysis of the cheeses that would enrich the work and support its commercial development. The Authors of the paper repeatedly highlighted the importance of sensory characteristics for consumers. Information on how changes in the profile of volatile compounds and fatty acids during cheese ripening have influenced their sensory characteristics is crucial for the acceptability of the product among consumers.
Response:
Nowadays, ripening time is increasing to produce high quality cheese for target markets. Cheese ripening is basically about the breakdown of proteins, lipids and carbohydrates (acids and sugars) which releases flavour compounds and modifies cheese texture.To justify the ripening costs, the nutritional properties of Apuseni cheese during the ripening period were presented. Apuseni cheese is an authentic cheese from Transylvania, which combines nutritional and sensory properties of ripening cheeses appreciated worldwide. The study of Apuseni cheese during the 16 months of ripening, showed that in Apuseni cheese there are high quality fatty acids and volatile compounds that give it a specific aroma.
There is a decrease in saturated fatty acids, which are recognized as health risk factors, but also an increase in monounsaturated fatty acids known for human health because of their effect in lowering blood cholesterol, in particular the DHA. Also note the presence of polyunsaturated fatty acids which are appreciated for their anti-thrombogenetic effect. The formation of taste and aroma substances takes place from the S2 phase of ripening: lactic acid gives the cheeses a sour, pleasant taste when it is in small quantities. Salt favors the individual highlighting of different substances of taste and flavor, but also the flavors derived from fats associated with ripening cheeses result from the release of fatty acids by lipolysis and subsequent modification of fatty acids by microorganisms to other compounds. Protein hydrolysis products influence the taste and aroma of cheeses. Flavour and texture development are strongly dependent on: pH profile, composition, salting, temperature, humidity, experience.
From the point of view of the acceptability of Apuseni cheese, it is very appreciated, being a very well sold product (it is sold both after 4 months from ripening and after 16 months). The organoleptic and physico-chemical characteristics are influenced by all these changes highlighted in this manuscript.In addition, the results can serve as a basis for the development of comparative analysis tools and strategies aimed at improving the nutritional characteristics of cheese ripening. Also, studies of the volatile profile of cheese could be an interesting method for studying the incorporation of further advantages such as the inclusion of certain commercial starters or the application of other treatments to enhance the quality or the shelf-life of cheeses.
We are grateful for your competent suggestions! Thank you for your patience and time!
Cluj-Napoca, 29.12.2020
Yours sincerely,
Mureşan et al.

Reviewer 2 Report
The manuscript entitled “Changes in physicochemical and microbiological properties, fatty acid and volatile compound profiles of Apuseni cheese during ripening” (foods-1057596) aimed to identify changes in chemical and microbiological properties and safety, fatty acid composition and volatile compounds profile of Apuseni cheese during ripening, up to 16 months, to improve knowledge about the quality of ripening cheese.
Generally, in my opinion, the manuscript needs much improvement and explanations, which should be addressed before considering for publication. Also, I strongly recommend giving the whole manuscript to native speaker, extensive editing of English language and style required.
There are some comments and questions to the Authors:
Abstract should be rewritten, including other comments. Try to focus more on obtained results.
Keywords should be redrafted. The main role of Keywords is to significantly expand further searching potential of the manuscript (please avoid using the same as in the title).
Introduction section is poorly written, does not include sufficient background of conducted research. Please also avoid general and well-known statements. Maybe Authors can focus more on the description of Apuseni cheese, the origin, nutritional value, safety, type of milk used for production, aroma compounds previously detected in this type of cheese, specific attributes, usual maturation period, etc.
Aim of study should be carefully redrafted. I encourage Authors to create hypothesis and focus more on the novelty of the study.
In materials and methods section please start with proper and detailed information of Sampling and Cheese manufacturing. For sampling add information about e.g.: number of cheeses, batches – to give representative data Authors should analyzed at least three batches from e.g., 3 or more cheeses randomly sampled in each batch.
Provide other description of storage conditions during ripening period - humidity, Authors stated that samples were taken from a milk processing unit in Romania (only fresh or during ripening also?), but in my opinion those information are important for experiment, and Authors need to remember that experiment should be well explained to the readers and give the opportunity to repeat the experiment based on the description.
For cheese manufacture add more specific information: when was manufactured the cheese samples, where exactly, by which procedure (include citation if possible), from where milk was obtained, what about the quality and safety of raw milk and hygiene conditions,
In my opinion, “Cheesemaking” description should be better detailed and may be given in table for better understanding.
In my opinion signs S1-S5 are not needed and a bit confusing. I recommend to the Authors simply use fresh samples (e.g., 0 months) etc. up to 16 months. In the text and in tables/ figures. Same for D1-D5.
Instead of physicochemical analysis Authors should name it as for e.g., proximate composition.
Line 95: “on a three-monthly basis during” explain, when S1 is 0 months, S2 is 4 months, not 3.
Authors provided many microbiological analyses, but please explain what about Listeria monocytogenes and e.g., Campylobacter spp.? Having in mind health safety of the cheese samples why Authors decided to check presence of those chosen pathogens.
Line 155: abbreviation of total lipids in my opinion is not needed, but I suggest including abbreviation for fatty acids (FAs), also for SFAs, MUFAs, PUFAs. Remember about space between 10 g; line 158: “[25].The”; line 166: unnecessary dot before “and”; line 168: delete one additional dot at the end of the sentence.
Why lipids were extracted by the method applied for tomatoes processing by-products? Do Authors think it is an appropriate method. What about e.g., Weibull-Stoldt method? Please explain.
Line 154: why not simply as follows: Determination of fatty acid. Composition in cheese.
Line 169: why not simply as follows: Analysis of volatile compounds in cheese.
Line 172: hyphenated? Better combined/ coupled.
Line 173: why in 60*C? Add proper citation of method used.
Line 178: GC-MS, correct.
Line 181: correct 40*C etc.
Line 187: instead of concentration use amount or content.
Line 188: abbreviation TIC is not needed. Not better as follows: were quantified and expressed as percentages of the total peak area?
Lines 190-191: this part should be completed as follows (add information needed in the text):
“All analyses were performed in triplicate. Differences in …. were analyzed using one-way analysis of variance ANOVA (or another method). Significance of differences between means for each parameter was determined by the Tukey’s test at a significance level of p < 0.05. All results were presented as mean ± SD. Include information about used software”.
Line 194: harvest is word more appropriate for fruits e.g., do not use expression “organoleptic” (it is incorrect) it is sensory characteristics/properties. Lines 194-198 are more appropriate for materials and methods section. Fig. 1 or delete or put in good quality and with better presentation, reconsider if it is really needed.
Proximate analysis results should be better explained and discussed with proper literature data published previously. In present form is too general.
With regard to all Tables include letters of significance after SD.
Line 228: legislation? Specify.
Line 232: better do not use this tense in scientific text, simple increased.
Line 234: what about any more recent limitations (if available).
Expression of pathogens counting should be better presented: I suggest simply as e.g. 4 x 10(5 as superscript) cfu/mL.
Line 253: secondary microflora? Specify.
Line 254: increase? Add more information like e.g., 2-fold more etc.
Table 3: better “not detected” instead of absent (depends on the method applied).
Lines 261-271: very general description, same as shown in table, correct, this should be comparison with other previous data and discussion of obtained results including proper explanations.
Line 279: source? Confront with nutritional standards.
Lines 282-284: use C16:0 instead of name of FAs, etc.; too little discussed. Please correct.
Title of table 4 should be simple as: fatty acid composition (%) of Apuseni cheese samples (it is known from materials and methods that by GC-MS). Names of FAs under the table are not needed.
Please explain why ripening period was analyzed up to 16 months, what was the limit? Why Authors decided to check only in 4 months intervals, what about changes between these months? What about safety and sensory analysis of samples for 16 months? Sensory is very important factor of quality of cheese.
If Authors decide to leave fig. 2 under the fig. Authors should include proper identification of FAs with explanation of each number 1-14.
Lines 301-307: not needed here, transfer to materials and methods section and correct.
Then should be more like: The relative abundance of volatile compounds in the Apuseni cheese samples, including ketones (5 compounds), alcohols (2), terpene hydrocarbons… (8), esters (5), aldehyde (1) and others (3) is given in Table 5.
Line 337: insert citation.
Line 338: With regard…
Change title of table 5 as Volatile compounds (%) identified in Apuseni cheese samples at five different stages of ripening (means +- SD, n = 3). How many repetitions? N=3?
In table 3 statistical analysis is missing, same for SD, like this is unacceptable. For volatiles analysis at least 3 repetitions are mandatory.
In results and discussion section I do not really see the discussion with literature data. Please try to confront obtained results with previous studies.
Please rewrite conclusions section, if possible, add some numerical results.
References: Please see the “Guide for Authors” and check carefully all literature items.
Author Response
Ms. Ref. No.: Foods- 1057596
Title: Changes in physicochemical and microbiological properties, fatty acid and volatile compound profiles of Apuseni cheese during ripening
Foods –MDPI – Section “Dairy” Special Issue "Functional and Nutritional Properties of Different Kinds of Milk"
On behalf of all co-authors,
I would like to thank you for taking necessary steps to improve our Manuscript in term of scientific quality. I also thank all the reviewers for spending their valuable time to correct our MS entitled “CHANGES IN PHYSICOCHEMICAL AND MICROBIOLOGICAL PROPERTIES, FATTY ACID AND VOLATILE COMPOUND PROFILES OF APUSENI CHEESE DURING RIPENING”. Based on your and the Reviewers’ valuable comments and suggestions, we revised our MS and we hope, the changes made in the revised MS are satisfactory.
All changes are marked as highlighted red letters in the revised MS.
Thank you for your confidence – according to your suggestions, we had performed amendments to our MS in order to better highlight its scientific potential.
Comments from Reviewer 2
The manuscript entitled “Changes in physicochemical and microbiological properties, fatty acid and volatile compound profiles of Apuseni cheese during ripening” (foods-1057596) aimed to identify changes in chemical and microbiological properties and safety, fatty acid composition and volatile compounds profile of Apuseni cheese during ripening, up to 16 months, to improve knowledge about the quality of ripening cheese.
Generally, in my opinion, the manuscript needs much improvement and explanations, which should be addressed before considering for publication. Also, I strongly recommend giving the whole manuscript to native speaker, extensive editing of English language and style required.
There are some comments and questions to the Authors:
Abstract should be rewritten, including other comments. Try to focus more on obtained results.
Response:
New Abstract: Cheese is one of the oldest foods, being known over 4000 types, appreciated for its nutritional, sensory and high quality fatty acid properties. Cheese ripening is basically about the breakdown of proteins, lipids and carbohydrates (acids and sugars) which releases flavour compounds and modifies cheese texture. The formation of taste and aroma substances takes place from the S2 phase of ripening: lactic acid gives the cheeses a sour, pleasant taste when it is in small quantities. The sour taste gradually disappears after the breakdown of lactic acid and is replaced by a taste similar to that of walnut kernels. The salt favors the individual highlighting of the different substances of taste and aroma, but also the fat that loosens the cheese paste and has an emulsifying effect. The fat derived flavors associated with cheese ripening result from the release of fatty acids by lipolysis and further modification of fatty acids by microorganisms to other compounds. Protein hydrolysis products influence the taste and aroma of cheeses, all the more intensely as a further decomposition has taken place. In the case of strongly ripening cheeses, as in the case of Apuseni Cheese, the presence of amines and ammonia contributes to the formation of the characteristic taste and aromaThe purpose of this study is to analyze the influence of ripening time (16 months) on the chemical properties, fatty acids and volatile componds of Apuseni cheese to improve knowledge about the quality of ripening cheese. As the cheese matures, it notices a decrease in sample moisture from 54.66% to 46.69% and pH from 5.44 to 5.19, at the same time an increase in the content of fat from 24.40% to 32.16%, respectively fat in to dry matter from 43.79% to 60.32% protein from 25.00% to 29.47%, ash from 4.04% to 4.91% and NaCl from 1.81% to 2.43. Even if the microbiological parameters increase during the 16 months of ripening, they fall within the maximum allowed limits. During ripening, 14 fatty acids were identified, with a decrease in saturated fatty acids from 77.93% to 68.42% and an increase in monounsaturated fatty acids from 20.82% to 28.98. The volatile compounds identified in the cheese samples are 2 alcohols, 1 aldehyde, 5 ketones, 8 hydrocarbon terpenes and oxygenated derivatives, as well as 5 esters and 3 other classes of compounds.
Keywords should be redrafted. The main role of Keywords is to significantly expand further searching potential of the manuscript (please avoid using the same as in the title).
Response: matured cheese, nutrients, lipids, aroma compounds
Introduction section is poorly written, does not include sufficient background of conducted research. Please also avoid general and well-known statements. Maybe Authors can focus more on the description of Apuseni cheese, the origin, nutritional value, safety, type of milk used for production, aroma compounds previously detected in this type of cheese, specific attributes, usual maturation period, etc.
Response:
New Introduction: The name 'cheese' is used for products obtained from fresh or ripened coagulated milk [1]. The history of cheese dates back to the Middle East, more than 10,000 years ago, when goats and sheep were domesticated, being considered one of the oldest foods [2].
The assortment range of cheeses is very wide all over the world. In the beginning, cheese was produced to increase the shelf life of milk, but now cheese is purchased for its sensory and nutritional qualities [3].
Worldwide, there are currently about 4,000 varieties of cheese, most of which come from European countries. Cheeses are foods that are obtained by removing whey from the curd formed by the coagulation of whole, skimmed or partially skimmed milk, cream, buttermilk, or mixtures thereof, can be matured or fresh [4]. By concentrating fats in the curd obtained by precipitating casein, cheeses become a very good source of vitamins A, K, E, D, fat-soluble than milk [5].
Milk composition, the technological process, and storage conditions are heterogeneity factors that influence nutrients and their quality in ripening cheeses. The physical and aromatic characteristics of the cheese are directly influenced by these conditions, which are the decision factor in customer preferences. Cheese ripening is basically about the breakdown of proteins, lipids and carbohydrates (acids and sugars) which releases flavour compounds and modifies cheese texture. In the broadest terms there are three sources of cheese flavour:
Flavours present in the original cheese milk, such as natural butter fat flavour and feed flavour. Breakdown products of milk proteins, fats and sugars which are released by microbial enzymes, enzymes endogenous to milk, and enzyme additives. Flavour and texture development are strongly dependent on: pH profile, composition, salting, temperature, humidity, experience.
As a general rule factors which increase the rate of ripening increase the risk of off flavour development, and reduce the period of time when the cheese is saleable. Protein degradation during cheese curing is a directed process resulting in protein fragments with desirable flavours. Dairy fat is a wonderfully rich source of flavours, because it contains an extremely diverse selection of fatty acids.
Dairy fat without any ripening during cheese making is an important contributor to cheese flavour and texture. Fat also acts as a flavour reservoir, so hydrophobic (fat soluble) flavours derived from protein breakdown are stored in the fat and released during mastication in the mouth. The fat derived flavours associated with cheese ripening result from the release of fatty acids by lipolysis and further modification of fatty acids by microorganisms to other compounds.
Benzaldehyde and (E,E)-2,4-nonadienal are unsaturated aldehydes formed by oxidation of unsaturated fatty acids. Instead, acetaldehyde is formed via lactic acid fermentation. These compounds are formed during the secondary phase of the auto-oxidation process and give rise to off-flavors [6, 7, 8, 9]. The aroma of each type of ripening cheese differs and is influenced by the ripening time which can vary between a few weeks or several years. Once the cheese matures, the microflora changes from the initial sample, lysis of starter cells or the death of some cells and the development of a secondary microflora, adventitious nonstarter [8]. The processes that take place during ripening include lactate metabolism, citrate and residual lactose, but also proteolysis and lipolysis. Modification of these processes influences the development of fatty acid metabolism, volatile aromatic compounds and amino acid metabolism. The most important primary change in terms of complexity that occurs in most cheeses is proteolysis [6]. Following the cheese ripening processes, changes were observed in the physical, chemical, microbiological, rheological parameters, but also in the sensory properties [3, 7, 10, 11].
The aroma of semi-hard cheeses is formed during the ripening processes, following the biochemical reactions caused by the interaction of starter bacteria, enzymes from the milk and rennet, lipases, and secondary flora [12]. Different biochemical and chemical processes occur during the ripening period of the cheese proteolysis, lipolysis and fermentation, which give rise to the texture and aroma specific to each type of cheese. [5]. The final choice by the consumer is made according to the sensory properties, the aroma being considered an essential criterion [13].
Cheese is an easily digestible food rich in high quality nutrients such as protein, fatty acids, vitamins and minerals. Cheese fatty acids are one of the most important sources of bioactive substances [14].
A major influence in the aroma of the cheese has free fatty acids released by lipolysis, especially those with short and medium chain (C4:0-C8:0, C10:0-C14:0), and those with long chain (> 14 carbon atoms) have a low influence on cheese flavor because they have high perception thresholds. FFAs have also been shown to act on the substrate of several metabolic pathways, and produce flavor and aroma compounds, such as lactones, methyl ketones, alkanes, esters and secondary alcohols [5].
The aroma of the cheese is an important quality characteristic, being the decisive factor from the consumer's point of view. The specific aroma of a cheese is the result of a combination of volatile and non-volatile chemical compounds, which are formed during theripening process from milk fats, proteins and carbohydrates. These processes are followed by a series of catabolic side reactions, which are responsible for forming the flavor profile, unique to each cheese [12, 15].
Apuseni cheese is a hard cheese, originally from Coltesti-Apuseni, Romania, it is a royal delicacy balanced delicately between sweet and salty. It is recognized by the holes formed in the decomposition process of lactic acid and glutamic acid, when existing microorganisms produce carbon dioxide that remains trapped in the dense mass of cheese. The taste is specific, slightly sweet and is due to propionic ferments.
Aim of study should be carefully redrafted. I encourage Authors to create hypothesis and focus more on the novelty of the study.
Response:
New aim: This study aimed to analyze Apuseni cheese during 16 months of ripening. The evaluation of Changes in physicochemical and microbiological properties, fatty acid and volatile compound profiles of Apuseni cheese during ripening was aimed at improving knowledge about the quality of ripening cheese. The results can be used to improve storage parameters and to develop benchmarking tools and internships designed to improve the nutritional properties of matured ripening cheese.
In materials and methods section please start with proper and detailed information of Sampling and Cheese manufacturing. For sampling add information about e.g.: number of cheeses, batches – to give representative data Authors should analyzed at least three batches from e.g., 3 or more cheeses randomly sampled in each batch.
Provide other description of storage conditions during ripening period - humidity, Authors stated that samples were taken from a milk processing unit in Romania (only fresh or during ripening also?), but in my opinion those information are important for experiment, and Authors need to remember that experiment should be well explained to the readers and give the opportunity to repeat the experiment based on the description.
For cheese manufacture add more specific information: when was manufactured the cheese samples, where exactly, by which procedure (include citation if possible), from where milk was obtained, what about the quality and safety of raw milk and hygiene conditions,
In my opinion, “Cheesemaking” description should be better detailed and may be given in table for better understanding.
Response:
The samples of ‘Apuseni’ cheeses were taken from a milk processing unit in Romania (Coltesti dairy products).
The ripening of the samples took place over a period of 16 months. Sampling has been carried out over several periods, even in different seasons, in order to have continuity and to be able to tell the true quality of this type of hard paste cheese: the first sampling: S1- 0 months of ripening; second sampling: S2- 4 months of ripening; third sampling: S3- 8 months of ripening; fourth sampling: S4- 12 months of ripening; fifth sampling: S5- 16 months of ripening.
Cheesemaking: The supply of cow's milk is made exclusively from authorized farms within a radius of 30 kilometers from the Colţeşti-Alba-Romania area, which supply the best quality milk, according to European norms. The normalization of milk is done at 2.8 - 3.2% g. Standardized cow's milk was pasteurized at 65° C for 30 min.
The cultures used are frozen Cryofast cultures, coagulation power 1800 IMCU/g, produced by Company of Sacco System, Italy. The lyophilized culture Lactococcus lactis ssp. Lactis, Streptococcus thermophiles and CaCl2 were added, and in the cogulation stage at 32-34° C / for 35 min the microbial enzymatic clot (Rhizomucor miehei), produced by Ideal Still Exim Srl, 1.8 g / 100 l, for 30 - 35 minutes was added. This was followed by the cutting of the curd, the separation of whey, a slight heating to 40° C, mixing and a new separation of whey with the formation of the curd. Curds formed, pressed case was subjected to salting, drying and maturation ripening in storage at temperatures of 13-14°C / for 15 days and then 2-8°C for 16 months.
The shape of the blocks is parallelepiped, weight 7 kg, height 7-10 cm, length 25 cm, width 25 cm.
The salting process is wet, in saturated brine, and the pH is 5.02. Typically, the temperature of the brine and the air in the salting room varies between 8 and 16 ° C, the relative humidity of the air being 90-95%. The duration of salting is 24 hours, after which it takes 1-2 hours.
The ripening cheese is kept at a temperature of 2 - 8 ° C and a relative humidity of 85 - 90%. Periodically, the shell is washed with salt water, and if the shell is too thick, it is scraped. The turning prevents the cheese from deforming, ensuring a uniform maturation throughout the mass. The cheeses are turned by hand. Samples were taken from the same batch (batch 90), in all phases S1-S5. The sample was taken from the same blocks, with a scabbard.
In my opinion signs S1-S5 are not needed and a bit confusing. I recommend to the Authors simply use fresh samples (e.g., 0 months) etc. up to 16 months. In the text and in tables/ figures. Same for D1-D5.
Response : I corrected the whole manuscript.
Instead of physicochemical analysis Authors should name it as for e.g., proximate composition.
Response: corrected
Line 95: “on a three-monthly basis during” explain, when S1 is 0 months, S2 is 4 months, not 3.
Response: corrected
Authors provided many microbiological analyses, but please explain what about Listeria monocytogenes and e.g., Campylobacter spp.? Having in mind health safety of the cheese samples why Authors decided to check presence of those chosen pathogens.
Response:
The incidence of Campylobacter spp. and Listeria monocytogenes in cheese ripening obtained from pasteurised milk is low according to the literature. The presence of Campylobacter spp. in raw milk may be contributed to contamination during the milking process from the farm environment through feces or after milking due to poor hygienic conditions during storage and handling of milk.
The high incidence of Campylobacter spp. and Listeria monocytogenes can be due to the unhygienic condition applied during production, and storage cheese samples. But, in our study, the results obtained regarding the total viable number showed a high degree of hygiene, which determined us not to include in the analysis table the two food-borne pathogen bacteria.
Line 155: abbreviation of total lipids in my opinion is not needed, but I suggest including abbreviation for fatty acids (FAs), also for SFAs, MUFAs, PUFAs. Remember about space between 10 g; line 158: “[25].The”; line 166: unnecessary dot before “and”; line 168: delete one additional dot at the end of the sentence.
Response: corrected
Why lipids were extracted by the method applied for tomatoes processing by-products? Do Authors think it is an appropriate method. What about e.g., Weibull-Stoldt method? Please explain.
Response: This is the method used, adjusted for cheese samples. Use the same method for other tests, for example:
Dulf, F.; Oroian, I.; Vodnar, D.; Socaciu, C.; Pintea, A. Lipid Classes and Fatty Acid Regiodistribution in Triacylglycerols of Seed Oils of Two Sambucus Species (S. nigra L. and S. ebulus L.). Molecules, 2013, 18(10), 11768–11782. doi:10.3390/molecules181011768
Line 154: why not simply as follows: Determination of fatty acid. Composition in cheese.
Response: corrected
Line 169: why not simply as follows: Analysis of volatile compounds in cheese.
Response: corrected
Line 172: hyphenated? Better combined/ coupled.
Response: corrected
Line 173: why in 60*C? Add proper citation of method used.
Response: corrected. Referecnce [27]
Gan, H.H.; Yan, B.; Linforth, R.S.T.; Fisk, I.D. Development and validation of an APCI-MS/GC–MS approach for the classification and prediction of Cheddar cheese maturity. Food Chem. 2016, 190, 442–447. doi:10.1016/j.foodchem.2015.05.096
Line 178: GC-MS, correct.
Response: corrected
Line 181: correct 40*C etc.
Response: corrected
Line 187: instead of concentration use amount or content.
Response: corrected
Line 188: abbreviation TIC is not needed. Not better as follows: were quantified and expressed as percentages of the total peak area?
Response: corrected
Lines 190-191: this part should be completed as follows (add information needed in the text):
“All analyses were performed in triplicate. Differences in …. were analyzed using one-way analysis of variance ANOVA (or another method). Significance of differences between means for each parameter was determined by the Tukey’s test at a significance level of p < 0.05. All results were presented as mean ± SD. Include information about used software”.
Response:
All analyses were performed in duplicate, using MINITAB software. Differences were analyzed using one-way analysis of variance ANOVA (Analysis of Variance), General Linear Model, one-way ANOVA . Significance of differences between means for each parameter was determined by the Tukey’s test at a significance level of p < 0.05. All results were presented as mean ± SD.
Line 194: harvest is word more appropriate for fruits e.g., do not use expression “organoleptic” (it is incorrect) it is sensory characteristics/properties. Lines 194-198 are more appropriate for materials and methods section. Fig. 1 or delete or put in good quality and with better presentation, reconsider if it is really needed.
Response: corrected
Proximate analysis results should be better explained and discussed with proper literature data published previously. In present form is too general.
Response: corrected
With regard to all Tables include letters of significance after SD.
Response: corrected
Line 228: legislation? Specify.
Response: TYMC max. 300 cfu/m3; TVC max. 600 cfu/m3, of the Order of the Ministry of Health -976/1998 Available online: http://legislatie.just.ro/Public/DetaliiDocument/
Line 232: better do not use this tense in scientific text, simple increased.
Response: corrected
Line 234: what about any more recent limitations (if available).
Expression of pathogens counting should be better presented: I suggest simply as e.g. 4 x 10(5 as superscript) cfu/mL.
Response: The legislation in force is used - Order of the Ministry of Health -976/1998.
Line 253: secondary microflora? Specify.
Response: microbiological parameters
Line 254: increase? Add more information like e.g., 2-fold more etc.
Response: according to Table 3
Table 3: better “not detected” instead of absent (depends on the method applied).
Response: corrected
Lines 261-271: very general description, same as shown in table, correct, this should be comparison with other previous data and discussion of obtained results including proper explanations.
Response: It is very important to note that the amount of saturated fatty acids decreases,significant difference (p < 0.05), with ripening from 77.93% to 68.42%, are recognized as health risk factors, as well as polyunsaturated fatty acids from 1.72% to 1.11%. Among the PUFA’s are known components with anti-atheratogenic action like 18:2n – 6 belonging to the n – 6 PUFA’s class, and in the more important n – 3 PUFA’s class, components such as 18:3n – 3 which are appreciated for their anti-thrombogenetic effect [39].
While the amount of monounsaturated fatty acids increases, significant difference (p < 0.05), from 20.82% to 28.98% during the 16 months of ripening the cheese. MUFAs resulted very efficient in reducing the coronaric deseases risk. Indeed MUFA have been recognized as beneficial as the PUFA’s n – 3 class for human health because of their effect in lowering blood cholesterol, in particular the DHA [39]. It is noteworthy that the ratio between n-6 / n-3 decreases, significant difference (p <0.05), from 8.55 in D1- 0 months of ripening to 2.17 in D5- 16 months of ripening.
Line 279: source? Confront with nutritional standards.
Response: source 42
Lines 282-284: use C16:0 instead of name of FAs, etc.; too little discussed. Please correct.
Response: corrected
Title of table 4 should be simple as: fatty acid composition (%) of Apuseni cheese samples (it is known from materials and methods that by GC-MS). Names of FAs under the table are not needed.
Response: corrected
Please explain why ripening period was analyzed up to 16 months, what was the limit? Why Authors decided to check only in 4 months intervals, what about changes between these months? What about safety and sensory analysis of samples for 16 months? Sensory is very important factor of quality of cheese.
Response: These periods were chosen together with the factory engineers, because the Apuseni cheese is marketed after a minimum of 4 months of ripening up to a maximum of 16 months.
If Authors decide to leave fig. 2 under the fig. Authors should include proper identification of FAs with explanation of each number 1-14.
Response: corrected
Lines 301-307: not needed here, transfer to materials and methods section and correct.
Then should be more like: The relative abundance of volatile compounds in the Apuseni cheese samples, including ketones (5 compounds), alcohols (2), terpene hydrocarbons… (8), esters (5), aldehyde (1) and others (3) is given in Table 5.
Response: corrected
Line 337: insert citation.
Response: are presented in the following lines
Line 338: With regard…
Response: continuation of the previous line
Change title of table 5 as Volatile compounds (%) identified in Apuseni cheese samples at five different stages of ripening (means +- SD, n = 3). How many repetitions? N=3?
Response: corrected. Yes 3
In table 3 statistical analysis is missing, same for SD, like this is unacceptable. For volatiles analysis at least 3 repetitions are mandatory.
Response: corrected , table 5
In results and discussion section I do not really see the discussion with literature data. Please try to confront obtained results with previous studies.
Response: the manuscript has been improved
Please rewrite conclusions section, if possible, add some numerical results.
Response:
New conclusions: Nowadays, ripening time is increasing to produce high quality cheese for target markets. Cheese ripening is basically about the breakdown of proteins, lipids and carbohydrates (acids and sugars) which releases flavour compounds and modifies cheese texture.To justify the ripening costs, the nutritional properties of Apuseni cheese during the ripening period were presented. Apuseni cheese is an authentic cheese from Transylvania, which combines nutritional and sensory properties of ripening cheeses appreciated worldwide. The study of Apuseni cheese during the 16 months of ripening, showed that in Apuseni cheese there are high quality fatty acids and volatile compounds that give it a specific aroma.
There is a decrease in saturated fatty acids, which are recognized as health risk factors, but also an increase in monounsaturated fatty acids known for human health because of their effect in lowering blood cholesterol, in particular the DHA. Also note the presence of polyunsaturated fatty acids which are appreciated for their anti-thrombogenetic effect. The formation of taste and aroma substances takes place from the D2- 4 month of ripening: lactic acid gives the cheeses a sour, pleasant taste when it is in small quantities. Salt favors the individual highlighting of different substances of taste and flavor, but also the flavors derived from fats associated with ripening cheeses result from the release of fatty acids by lipolysis and subsequent modification of fatty acids by microorganisms to other compounds. Protein hydrolysis products influence the taste and aroma of cheeses. Flavour and texture development are strongly dependent on: pH profile, composition, salting, temperature, humidity, experience.
From the point of view of the acceptability of Apuseni cheese, it is very appreciated, being a very well sold product (it is sold both after 4 months from ripening and after 16 months). The organoleptic and physico-chemical characteristics are influenced by all these changes highlighted in this manuscript.In addition, the results can serve as a basis for the development of comparative analysis tools and strategies aimed at improving the nutritional characteristics of cheese ripening. Also, studies of the volatile profile of cheese could be an interesting method for studying the incorporation of further advantages such as the inclusion of certain commercial starters or the application of other treatments to enhance the quality or the shelf-life of cheeses.
References: Please see the “Guide for Authors” and check carefully all literature items.
Response: corrected
We are grateful for your competent suggestions! Thank you for your patience and time!
Cluj-Napoca, 01.01.2021
Yours sincerely,
Mureşan et al.
Reviewer 3 Report
Dear editor, the ms deals with an Apuseni cheese characterization during ripening by several indicators. The physicochemical properties analyzed were moisture, fat, ash, protein, NaCl, and pH. In addition microbiological analyses (e.g. Escherichia. coli, Staphylococcus aureus, and Salmonella), fatty acid composition, and volatile compounds were evaluated. The paper results interesting but it has some issues to be solved:
- the introduction is highly fragmented, should be more homogeneous;
- table 4, some results have been expressed at the second decimal place but the S.D. reached the third one, this is not scientifically correct;
- table 5, some compounds are reported as 0.00 what does it mean? Maybe the author would like to express not detected? this is very different
- The conclusions are too general and must be supported by the results obtained
Author Response
Ms. Ref. No.: Foods- 1057596
Title: Changes in physicochemical and microbiological properties, fatty acid and volatile compound profiles of Apuseni cheese during ripening
Foods –MDPI – Section “Dairy” Special Issue "Functional and Nutritional Properties of Different Kinds of Milk"
On behalf of all co-authors,
I would like to thank you for taking necessary steps to improve our Manuscript in term of scientific quality. I also thank all the reviewers for spending their valuable time to correct our MS entitled “CHANGES IN PHYSICOCHEMICAL AND MICROBIOLOGICAL PROPERTIES, FATTY ACID AND VOLATILE COMPOUND PROFILES OF APUSENI CHEESE DURING RIPENING”. Based on your and the Reviewers’ valuable comments and suggestions, we revised our MS and we hope, the changes made in the revised MS are satisfactory.
All changes are marked as highlighted red letters in the revised MS.
Thank you for your confidence – according to your suggestions, we had performed amendments to our MS in order to better highlight its scientific potential.
Reviewers' comments:
Dear editor, the ms deals with an Apuseni cheese characterization during ripening by several indicators. The physicochemical properties analyzed were moisture, fat, ash, protein, NaCl, and pH. In addition microbiological analyses (e.g. Escherichia. coli, Staphylococcus aureus, and Salmonella), fatty acid composition, and volatile compounds were evaluated. The paper results interesting but it has some issues to be solved:
- the introduction is highly fragmented, should be more homogeneous;
Response: New introduction
The name 'cheese' is used for products obtained from fresh or ripened coagulated milk [1]. The history of cheese dates back to the Middle East, more than 10,000 years ago, when goats and sheep were domesticated, being considered one of the oldest foods [2].
The assortment range of cheeses is very wide all over the world. In the beginning, cheese was produced to increase the shelf life of milk, but now cheese is purchased for its sensory and nutritional qualities [3].
Worldwide, there are currently about 4,000 varieties of cheese, most of which come from European countries. Cheeses are foods that are obtained by removing whey from the curd formed by the coagulation of whole, skimmed or partially skimmed milk, cream, buttermilk, or mixtures thereof, can be matured or fresh [4]. By concentrating fats in the curd obtained by precipitating casein, cheeses become a very good source of vitamins A, K, E, D, fat-soluble than milk [5].
Milk composition, the technological process, and storage conditions are heterogeneity factors that influence nutrients and their quality in ripening cheeses. The physical and aromatic characteristics of the cheese are directly influenced by these conditions, which are the decision factor in customer preferences. Cheese ripening is basically about the breakdown of proteins, lipids and carbohydrates (acids and sugars) which releases flavour compounds and modifies cheese texture. In the broadest terms there are three sources of cheese flavour:
Flavours present in the original cheese milk, such as natural butter fat flavour and feed flavour. Breakdown products of milk proteins, fats and sugars which are released by microbial enzymes, enzymes endogenous to milk, and enzyme additives. Flavour and texture development are strongly dependent on: pH profile, composition, salting, temperature, humidity, experience.
As a general rule factors which increase the rate of ripening increase the risk of off flavour development, and reduce the period of time when the cheese is saleable. Protein degradation during cheese curing is a directed process resulting in protein fragments with desirable flavours. Dairy fat is a wonderfully rich source of flavours, because it contains an extremely diverse selection of fatty acids.
Dairy fat without any ripening during cheese making is an important contributor to cheese flavour and texture. Fat also acts as a flavour reservoir, so hydrophobic (fat soluble) flavours derived from protein breakdown are stored in the fat and released during mastication in the mouth. The fat derived flavours associated with cheese ripening result from the release of fatty acids by lipolysis and further modification of fatty acids by microorganisms to other compounds.
Benzaldehyde and (E,E)-2,4-nonadienal are unsaturated aldehydes formed by oxidation of unsaturated fatty acids. Instead, acetaldehyde is formed via lactic acid fermentation. These compounds are formed during the secondary phase of the auto-oxidation process and give rise to off-flavors. [6, 7, 8, 9]. The aroma of each type of ripening cheese differs and is influenced by the ripening time which can vary between a few weeks or several years. Once the cheese matures, the microflora changes from the initial sample, lysis of starter cells or the death of some cells and the development of a secondary microflora, adventitious nonstarter [8]. The processes that take place during ripening include lactate metabolism, citrate and residual lactose, but also proteolysis and lipolysis. Modification of these processes influences the development of fatty acid metabolism, volatile aromatic compounds and amino acid metabolism. The most important primary change in terms of complexity that occurs in most cheeses is proteolysis [6]. Following the cheese ripening processes, changes were observed in the physical, chemical, microbiological, rheological parameters, but also in the sensory properties. [3, 7, 10, 11].
The aroma of semi-hard cheeses is formed during the ripening processes, following the biochemical reactions caused by the interaction of starter bacteria, enzymes from the milk and rennet, lipases, and secondary flora [12]. Different biochemical and chemical processes occur during the ripening period of the cheese proteolysis, lipolysis and fermentation, which give rise to the texture and aroma specific to each type of cheese. [5]. The final choice by the consumer is made according to the sensory properties, the aroma being considered an essential criterion [13].
Cheese is an easily digestible food rich in high quality nutrients such as protein, fatty acids, vitamins and minerals. Cheese fatty acids are one of the most important sources of bioactive substances [14].
A major influence in the aroma of the cheese has free fatty acids released by lipolysis, especially those with short and medium chain (C4:0-C8:0, C10:0-C14:0), and those with long chain (> 14 carbon atoms) have a low influence on cheese flavor because they have high perception thresholds. FFAs have also been shown to act on the substrate of several metabolic pathways, and produce flavor and aroma compounds, such as lactones, methyl ketones, alkanes, esters and secondary alcohols [5].
The aroma of the cheese is an important quality characteristic, being the decisive factor from the consumer's point of view. The specific aroma of a cheese is the result of a combination of volatile and non-volatile chemical compounds, which are formed during theripening process from milk fats, proteins and carbohydrates. These processes are followed by a series of catabolic side reactions, which are responsible for forming the flavor profile, unique to each cheese [12, 15].
Apuseni cheese is a hard cheese, it is a royal delicacy balanced delicately between sweet and salty. It is recognized by the holes formed in the decomposition process of lactic acid and glutamic acid, when existing microorganisms produce carbon dioxide that remains trapped in the dense mass of cheese. The taste is specific, slightly sweet and is due to propionic ferments.
This study aimed to identify changes in physicochemical and microbiological properties, fatty acid and volatile compound profiles of Apuseni cheese during ripening to improve knowledge about the quality of ripening cheese. The results can be used to develop benchmarking tools and internships designed to improve the nutritional properties of ripening cheese. Moreover, the volatile profile of ripening cheeses could be used to study the incorporation of certain compounds, the inclusion of certain commercial starters or the improvement of treatments to increase the quality or shelf life of cheeses.
- table 4, some results have been expressed at the second decimal place but the S.D. reached the third one, this is not scientifically correct;
Response: I passed all the results to 3 decimal places
- table 5, some compounds are reported as 0.00 what does it mean? Maybe the author would like to express not detected? this is very different
Response: Corrected
- The conclusions are too general and must be supported by the results obtained
Response: New conclusions
Nowadays, ripening time is increasing to produce high quality cheese for target markets. Cheese ripening is basically about the breakdown of proteins, lipids and carbohydrates (acids and sugars) which releases flavour compounds and modifies cheese texture.To justify the ripening costs, the nutritional properties of Apuseni cheese during the ripening period were presented. Apuseni cheese is an authentic cheese from Transylvania, which combines nutritional and sensory properties of ripening cheeses appreciated worldwide. The study of Apuseni cheese during the 16 months of ripening, showed that in Apuseni cheese there are high quality fatty acids and volatile compounds that give it a specific aroma.
There is a decrease in saturated fatty acids, which are recognized as health risk factors, but also an increase in monounsaturated fatty acids known for human health because of their effect in lowering blood cholesterol, in particular the DHA. Also note the presence of polyunsaturated fatty acids which are appreciated for their anti-thrombogenetic effect. The formation of taste and aroma substances takes place from the S2 phase of ripening: lactic acid gives the cheeses a sour, pleasant taste when it is in small quantities. Salt favors the individual highlighting of different substances of taste and flavor, but also the flavors derived from fats associated with ripening cheeses result from the release of fatty acids by lipolysis and subsequent modification of fatty acids by microorganisms to other compounds. Protein hydrolysis products influence the taste and aroma of cheeses. Flavour and texture development are strongly dependent on: pH profile, composition, salting, temperature, humidity, experience.
From the point of view of the acceptability of Apuseni cheese, it is very appreciated, being a very well sold product (it is sold both after 4 months from ripening and after 16 months). The organoleptic and physico-chemical characteristics are influenced by all these changes highlighted in this manuscript.In addition, the results can serve as a basis for the development of comparative analysis tools and strategies aimed at improving the nutritional characteristics of cheese ripening. Also, studies of the volatile profile of cheese could be an interesting method for studying the incorporation of further advantages such as the inclusion of certain commercial starters or the application of other treatments to enhance the quality or the shelf-life of cheeses.
We are grateful for your competent suggestions! Thank you for your patience and time!
Cluj-Napoca, 29.12.2020
Yours sincerely,
Mureşan et al.

Round 2
Reviewer 1 Report
Review of the paper: Changes in physicochemical and microbiological properties, fatty acid and volatile compound profiles of Apuseni cheese during ripening.
I am sorry to say this, but after the Authors' corrections, the manuscript has become even more chaotic and its scientific value is lower than in the original version.
Abstract: Authors should focus on the results obtained, not copy an introduction.
Introduction: The information provided is not sufficient to provide a background to the research undertaken.
Information about Apuseni cheese is superficial. The authors emphasize in the introduction the uniqueness of the taste of this cheese but completely ignore the suggestion to present the results of sensory evaluation of cheese during ripening.
The assumptions of the statistical analysis are not clear.
Methodology: still lacks information on the number of batches of cheeses tested, the number of samples. The description of the technological process is very chaotic. Based on the description of the methodology it is not possible to repeat such an experiment.
The discussion of the results and discussion is very poor, the conclusion does not present any information that could be relevant for consumers or cheese producers.
Author Response
Ms. Ref. No.: Foods- 1057596
Title: Changes in physicochemical and microbiological properties, fatty acid and volatile compound profiles of Apuseni cheese during ripening
Foods –MDPI – Section “Dairy” Special Issue "Functional and Nutritional Properties of Different Kinds of Milk"
On behalf of all co-authors,
I would like to thank you for taking necessary steps to improve our Manuscript in term of scientific quality. I also thank all the reviewers for spending their valuable time to correct our MS entitled “CHANGES IN PHYSICOCHEMICAL AND MICROBIOLOGICAL PROPERTIES, FATTY ACID AND VOLATILE COMPOUND PROFILES OF APUSENI CHEESE DURING RIPENING”. Based on your and the Reviewers’ valuable comments and suggestions, we revised our MS and we hope, the changes made in the revised MS are satisfactory.
All changes are marked as highlighted red letters in the revised MS.
Thank you for your confidence – according to your suggestions, we had performed amendments to our MS in order to better highlight its scientific potential.
Reviewers' comments:
Reviewer 1
I am sorry to say this, but after the Authors' corrections, the manuscript has become even more chaotic and its scientific value is lower than in the original version.
Response:
In the first revision I tried to respond punctually to each comment of each reviewer. I revised the entire manuscript. The entire revised manuscript can be found below.
Abstract: Authors should focus on the results obtained, not copy an introduction.
Response: I made a new abstract.
Introduction: The information provided is not sufficient to provide a background to the research undertaken.
Response: I made a new introduction
Information about Apuseni cheese is superficial. The authors emphasize in the introduction the uniqueness of the taste of this cheese but completely ignore the suggestion to present the results of sensory evaluation of cheese during ripening.
Response: I made a new detailed description about Apuseni cheese.
Regarding the sensory analysis, it was not in the initial design of the project, and now it is too late because we have no more evidence to do the sensory analysis.
The assumptions of the statistical analysis are not clear.
Response: Corrected - for samples D1-D5
Methodology: still lacks information on the number of batches of cheeses tested, the number of samples. The description of the technological process is very chaotic. Based on the description of the methodology it is not possible to repeat such an experiment.
Response: In the manuscript we have detailed both the technological process with the help of a flow of the diagram, in order to offer all the details as easy as possible to understand, as well as about the samples used.
For this experiment, samples were collected from a single batch, in order to see the changes that take place during 16 months of ripening, using the same raw material. I did not use several batches because the chemical composition of the milk changes from one season to another. The lot has 47 blocks. According to the standard SR 13438: 1999 Milk and dairy products. Rules for quality verification (https://magazin.asro.ro/ro/standard/24527). Samples were collected from 5 different blocks with a scabbard, at each collection, from the same batch, which were homogenized and analyzed in duplicate.
The discussion of the results and discussion is very poor, the conclusion does not present any information that could be relevant for consumers or cheese producers.
Response: The results, discussions and conclusions have been improved
We are grateful for your competent suggestions! Thank you for your patience and time!
Cluj-Napoca, 11.01.2021
Yours sincerely,
Mureşan et al.

Reviewer 2 Report
In my opinion the revised version of manuscript foods-1057596 still needs much improvement, which should be addressed before reconsidering for its suitability for publication. Authors revised manuscript accordingly not to all of Reviewers comments and suggestions, and did not correct manuscript correctly. Also, there is the same problem, as I mentioned in my first review form, extensive editing of English language and style required. In many places the style is very confusing.
Generally, I do appreciate all the effort made by the Authors during preparing revision version of the manuscript, but in present form manuscript is not suitable for publication.
Please see review report forms again and revise manuscript with regard to all comments and suggestions. Also, I recommend to the Authors try to underline novelty, hypothesis and thoughout the experiment and results obtained. The presented topic of research is quite interesting and worth further investigation.
Please see some more detailed suggestions:
All manuscript should be properly redrafted. In abstract please focus more on obtained results (please check, but as I see abstract section after revision exceeded word limit); also, in introduction Authors should present more detailed description of Apuseni cheese, as was mentioned earlier etc.
I strongly recommend to carefully redraft all materials and method section, with special attention to sampling and cheese manufacture. Maybe reconsider including some table for better understanding and dividing for some subtitles.
"Samples were taken from the same batch (batch 90), in all phases D1-D5. The sample was taken from the same blocks, with a scabbard" - presented correction is not very informative. Also, it is not the proper answer for Reviewer comments. One batch is not enough to conduct such experiment, and in my opinion is unacceptable. Please see some more previously published papers by other Authors, this may be very helpful.
Abbreviations as D1-D5 instead of 0-16 months, in my opinion is not appropriate.
Fig. 1 should be of better quality and presentation. Why doubled fig.1?
Expression of FAs would be enough with decimal places, as thousandths place in my opinion is not needed.
In case of volatiles better use not identified or detected instead of "-".
Author Response
Ms. Ref. No.: Foods- 1057596
Title: Changes in physicochemical and microbiological properties, fatty acid and volatile compound profiles of Apuseni cheese during ripening
Foods –MDPI – Section “Dairy” Special Issue "Functional and Nutritional Properties of Different Kinds of Milk"
On behalf of all co-authors,
I would like to thank you for taking necessary steps to improve our Manuscript in term of scientific quality. I also thank all the reviewers for spending their valuable time to correct our MS entitled “CHANGES IN PHYSICOCHEMICAL AND MICROBIOLOGICAL PROPERTIES, FATTY ACID AND VOLATILE COMPOUND PROFILES OF APUSENI CHEESE DURING RIPENING”. Based on your and the Reviewers’ valuable comments and suggestions, we revised our MS and we hope, the changes made in the revised MS are satisfactory.
All changes are marked as highlighted red letters in the revised MS.
Thank you for your confidence – according to your suggestions, we had performed amendments to our MS in order to better highlight its scientific potential.
Reviewers' comments:
Reviewer 2
In my opinion the revised version of manuscript foods-1057596 still needs much improvement, which should be addressed before reconsidering for its suitability for publication. Authors revised manuscript accordingly not to all of Reviewers comments and suggestions, and did not correct manuscript correctly. Also, there is the same problem, as I mentioned in my first review form, extensive editing of English language and style required. In many places the style is very confusing.
Generally, I do appreciate all the effort made by the Authors during preparing revision version of the manuscript, but in present form manuscript is not suitable for publication.
Please see review report forms again and revise manuscript with regard to all comments and suggestions. Also, I recommend to the Authors try to underline novelty, hypothesis and thoughout the experiment and results obtained. The presented topic of research is quite interesting and worth further investigation.
Please see some more detailed suggestions:
All manuscript should be properly redrafted. In abstract please focus more on obtained results (please check, but as I see abstract section after revision exceeded word limit); also, in introduction Authors should present more detailed description of Apuseni cheese, as was mentioned earlier etc.
Response: The abstract and introduction have been improved. The whole manuscript has been improved.
I strongly recommend to carefully redraft all materials and method section, with special attention to sampling and cheese manufacture. Maybe reconsider including some table for better understanding and dividing for some subtitles.
Response: In the manuscript we have detailed both the technological process with the help of a flow of the diagram, in order to offer all the details as easy as possible to understand, as well as about the samples used.
"Samples were taken from the same batch (batch 90), in all phases D1-D5. The sample was taken from the same blocks, with a scabbard" - presented correction is not very informative. Also, it is not the proper answer for Reviewer comments. One batch is not enough to conduct such experiment, and in my opinion is unacceptable. Please see some more previously published papers by other Authors, this may be very helpful.
Response: For this experiment, samples were collected from a single batch, in order to see the changes that take place during 16 months of ripening, using the same raw material. I did not use several batches because the chemical composition of the milk changes from one season to another. The lot has 47 blocks. According to the standard SR 13438: 1999 Milk and dairy products. Rules for quality verification (https://magazin.asro.ro/ro/standard/24527). Samples were collected from 5 different blocks with a scabbard, at each collection, from the same batch, which were homogenized and analyzed in duplicate.
Abbreviations as D1-D5 instead of 0-16 months, in my opinion is not appropriate.
Response: I used these abbreviations on the recommendation of another reviewer
Fig. 1 should be of better quality and presentation. Why doubled fig.1?
Response: I changed Figure 1.
Figure 1 has not been duplicated. It was moved to Material and because I used track changes it was just cut
Expression of FAs would be enough with decimal places, as thousandths place in my opinion is not needed.
Response: At the recommendation of another reviewer I modified (he said that it is not scientific to have 2 decimals in the results and 3 in the SD)
In case of volatiles better use not identified or detected instead of "-".
Response: Corrected
We are grateful for your competent suggestions! Thank you for your patience and time!
Cluj-Napoca, 11.01.2021
Yours sincerely,
Mureşan et al.

Reviewer 3 Report
Dear editor,
the authors satisfied all my queries.
Nevertheless, in the paper rewriting the abstract has exceeded the maximum word allowed. It must be shortened.
Author Response
Ms. Ref. No.: Foods- 1057596
Title: Changes in physicochemical and microbiological properties, fatty acid and volatile compound profiles of Apuseni cheese during ripening
Foods –MDPI – Section “Dairy” Special Issue "Functional and Nutritional Properties of Different Kinds of Milk"
On behalf of all co-authors,
I would like to thank you for taking necessary steps to improve our Manuscript in term of scientific quality. I also thank all the reviewers for spending their valuable time to correct our MS entitled “CHANGES IN PHYSICOCHEMICAL AND MICROBIOLOGICAL PROPERTIES, FATTY ACID AND VOLATILE COMPOUND PROFILES OF APUSENI CHEESE DURING RIPENING”. Based on your and the Reviewers’ valuable comments and suggestions, we revised our MS and we hope, the changes made in the revised MS are satisfactory.
All changes are marked as highlighted red letters in the revised MS.
Thank you for your confidence – according to your suggestions, we had performed amendments to our MS in order to better highlight its scientific potential.
Reviewers' comments:
the authors satisfied all my queries.
Nevertheless, in the paper rewriting the abstract has exceeded the maximum word allowed. It must be shortened.
Response: corrected
The evolution during ripening on the quality of Apuseni cheese was studied in this research. The cheese samples were controled and evaluated periodically (at 4 months) during 16 months of storage (at 2-8°C) for physicochemical parameters (pH, moisture, fat, fat in dry matter, total protein, ash , NaCl), microbiological (total combined yeasts and molds count (TYMC), total viable count (TVC), Escherichia coli, Staphylococcus aureus, Salmonella, lactic acid bacteria (LAB)), fatty acids (FA) and volatile compounds. For a better control on the quality of the cheese, the storage space was evaluated for TYMC and TVC. The ripening period showed improved effects on the quality of the cheese, showing lower values for moisture and pH and an increase in macronutrients. Both the cheese samples and the storage space were kept within the allowed microbiological limits. Lipids are predominant, the predominant FAs being saturated fatty acids (SFAs) which decrease, while monounsaturated fatty acids (MUFAs) increase. During ripening the microbiological and chemical changes result in the development of flavor. Major volatile compounds such as 2-Heptanone show accumulations, while Acetophenone, Limonene or Thymol show a decrease. In conclusion, Apuseni ripening cheese clearly involves a complex series of transformations, leading to a ripening cheese with improved nutritional and aromatic characteristics.
We are grateful for your competent suggestions! Thank you for your patience and time!
Cluj-Napoca, 11.01.2021
Yours sincerely,
Mureşan et al.
